# ZSMERGE: ZERO-SHOT KV CACHE COMPRESSION FOR MEMORY-EFFICIENT LONG-CONTEXT LLMS

## ABSTRACT

The linear growth of key-value (KV) cache memory and quadratic computational complexity in attention mechanisms pose significant bottlenecks for large language models (LLMs) in long-context processing. While existing KV cache optimization methods address these challenges through token pruning or feature merging, they often incur irreversible information loss or require costly retraining. To this end, we propose ZSMerge, a dynamic KV cache compression framework for efficient cache management, featuring three key operations: (1) fine-grained memory allocation guided by multi-dimensional token importance metrics at head-level granularity, (2) a residual merging mechanism that preserves critical context through compensated attention scoring, and (3) a zero-shot adaptation mechanism compatible with diverse LLM architectures without retraining. ZSMerge significantly enhances memory efficiency and inference speed. When applied to LLaMA2-7B, it achieves a 20:1 compression ratio for KV cache retention (reducing memory footprint to 5% of baseline) while sustaining generation quality and achieving a 2.25× throughput improvement at extreme 54k-token contexts, eliminating out-of-memory failures. These gains extend across diverse LLM families and long-context benchmarks, with ZSMerge consistently surpassing state-of-the-art cache optimization methods across tasks such as summarization, QA, reasoning, and code completion. The code is available at `https://anonymous.4open.science/r/ZSMerge-FC36`.

## 1 INTRODUCTION

The advancement of the Transformer architecture has revolutionized sequence data processing, with Large Language Models (LLMs) being a prime illustration (Vaswani et al., 2017). LLMs have achieved remarkable progress, surpassing human performance in diverse applications. While multi-turn dialogue and long-context understanding are critical scenarios (Yi et al., 2024; Gao et al., 2023; Li et al., 2023; Liu et al., 2024), handling extended sequences remains challenging due to the linear expansion of the Key-Value (KV) cache, which stores intermediate attention keys and values during generation to avoid redundant computations (Kwon et al., 2023). For instance, deploying a LLaMA2-7B (Sarah et al., 2024) model will consume about 26GB of GPU memory; when the token length reaches 32K, the KV cache alone occupies 32GB of memory, becoming the primary memory consumption. This, along with the quadratic computational complexity, significantly restricts LLM integration and performance, thus motivating recent research efforts toward optimizing KV cache utilization for enhanced inference efficiency (Zhang et al., 2024b; Sun et al., 2024; Wang et al., 2024).

To mitigate the large memory footprint of the KV cache, various optimization strategies have emerged. Some modify the core model architecture (Sarah et al., 2024; Ashkboos et al., 2024) or employ quantization techniques for lower precision representation (Dettmers et al., 2022; Xiao et al., 2023a). Other methods directly target the KV cache bottleneck during inference using context-aware techniques. Common among these is *sparse approximation* or *token eviction*, retaining only crucial tokens while discarding others (Zhang et al., 2024e;c); however, this can disrupt KV embeddings and cause information loss. To mitigate this, alternative context-aware techniques *merge* or *fuse* information from similar tokens (Dong et al., 2024), often using low-rank compression, though these may introduce architectural overhead or require model retraining.

Ideally, an effective and practical KV cache management strategy should 1) strictly control the memory consumption to break the context length limits, 2) minimize the generation quality degrada-

Figure 1: Comparative analysis of contextually adaptive KV cache optimization strategies. (A) **Sparse Approximation (Eviction)**: Low-contribution tokens are permanently discarded, reducing memory but risking error propagation. (B) **Token Merging**: Similar tokens are fused to mitigate information loss, but may require fine-tuning or auxiliary networks. (C) **ZSMerge (Proposed)**: Achieves zero-shot "sparse+residual" compression without added parameters or tuning, balancing efficiency and performance.

tion, and 3) readily applicable across diverse LLM backbones without necessitating fine-tuning or architectural changes (zero-shot compatibility).

Motivated by these goals, we propose ZSMerge, an efficient, comprehensive, and dynamic zero-shot KV management algorithm, with the following key features:

- **Fine-grained memory allocation**: ZSMerge leverages multi-dimensional importance metrics: exponential-decay scoring for temporal significance, head-level budget allocation for spatial adaptation, and similarity-based merging for semantic coherence (detailed in Section 3). This approach maintains superior generation quality compared to sparse eviction methods while achieving performance comparable to retraining-dependent approaches, even under significantly reduced memory budgets.

- **Zero-shot and low-cost Integration**: Designed for ease of use, ZSMerge does not introduce any model parameters and thus requires no retraining or model fine-tuning to adapt to different compression ratios or various task types. This also makes it compatible to various mainstream LLM architectures such as LLaMA, Falcon (Almazrouei et al., 2023), Mistral (Jiang et al., 2023), Qwen (Yang et al., 2024), and Yi (AI et al.), thereby minimizing deployment overhead.

- **Efficient throughput-boosting strategy**: Characterized by linear operational complexity, ZSMerge's compression mechanism is computationally lightweight. This efficiency brings significant throughput improvement, e.g., it helps achieve a 2.25× inference throughput improvement compared to the original model at a token length of 54K. Furthermore, we have proved that the KV cache compressed by ZSMerge can effectively preserves the informational contribution of retained tokens, preventing signal degradation even as context length increases substantially.

## 2 RELATED WORK

KV cache growth in long-context LLMs creates severe memory and computational bottlenecks, motivating optimization strategies across two categories: context-agnostic and contextually adaptive approaches.

### 2.1 CONTEXT-AGNOSTIC OPTIMIZATION

Context-agnostic methods reduce resource consumption through structural or numerical modifications independent of input sequences. Key approaches include: (1) **Structural compression** via knowledge distillation (Gu et al., 2024) or matrix factorization (Ashkboos et al., 2024); (2) **Attention optimization** through head pruning (Voita et al., 2019) and key-value sharing in Multi-Query (Shazeer, 2019) and Grouped-Query Attention (Ainslie et al., 2023); (3) **Numerical optimization** via post-training

quantization (Frantar et al., 2022), hardware-aligned quantization (Chen et al., 2024), and low-rank approximation (Wang et al., 2020).

## 2.2 CONTEXTUALLY ADAPTIVE OPTIMIZATION

Context-aware methods directly target KV cache bottlenecks by exploiting attention sparsity (Ji et al., 2021; Deng et al., 2024). Figure 1 illustrates three main approaches:

**Sparse Approximation (Eviction):** Methods directly reduce memory by permanently discarding low-contribution tokens. Xiao et al. (2023b) retains "attention sinks" with sliding windows, while Han et al. (2024) uses specialized masks for local-global balance. Li et al. (2024) performs static prefilling-stage pruning, Zhang et al. (2024e) employs attention score thresholds, and Zhang et al. (2024c) uses layer-wise patterns. Recent work includes prefilling-specific sparse attention (Lai et al.) and adaptive eviction patterns (Ge et al., 2023; Roy et al., 2021). Emerging techniques employ diverse strategies to address long-context challenges: retrieval-aware approaches (Xiao et al., 2024), query-agnostic compression (Kim et al., 2025), and dual-stage compression (Behnam et al., 2025). *Limitation*: Permanent token removal causes attention distribution drift and error propagation (Dao et al., 2022; Xiao et al., 2023b).

**Token Merging:** Methods fuse similar tokens to mitigate information loss from eviction. Token merging was pioneered in computer vision by Bolya et al. (2022), which merges spatial tokens in vision transformers for computational efficiency. Adapting this concept to LLM KV cache compression, Nawrot et al. (2024) adaptively merges spatial tokens with fine-tuning, while Dong et al. (2024) uses auxiliary networks for minimal attention impact. Wu et al. (2025) and Zhang et al. (2024d) employ attention-weight-based adaptive merging with merge ratios proportional to attention scores to directly minimize output distortion. Xu et al. applies orthogonal query-driven channel pruning. *Limitation*: Existing merging approaches typically either rely on auxiliary modules for fitting or lack theoretical guarantees for attention distribution preservation, complicating deployment and obscuring actual performance sources.

**Advanced Hybrid Approaches:** Methods combine multiple compression techniques: MLP-based compressor modules co-trained with attention (Yuan et al.), block-level pruning with auxiliary routing networks for gating (MoBA) (Lu et al.), dynamic context selection and offloading (Hao et al.), fixed budget periodic compression (Kim et al.), constant-sized KV caches for extended responses (Ghadia et al., 2025), and KV cache-centric analysis frameworks (Li et al., 2025). *Limitation*: Require retraining, architectural changes, or specialized system configurations, complicating integration and obscuring performance attribution.

**ZSMerge (Proposed):** Addressing these limitations, ZSMerge combines sparse approximation with residual merging in a zero-shot framework. Unlike eviction methods, it preserves information through compensated attention scoring to prevent error propagation. Unlike token merging, it requires no auxiliary networks or fine-tuning. Unlike hybrid approaches, it demands no retraining or architectural changes, enabling direct integration into existing LLM inference pipelines while maintaining clear performance attribution.

## 3 ZSMERGE METHODOLOGY

Building upon the conceptual foundation of ZSMerge outlined earlier, this section delineates the technical framework underpinning its zero-shot compression mechanism. We propose a four-component methodology consisting of adaptive budget allocation, context-sensitive contribution evaluation, residual token merging, and stabilized attention projection. The interplay of these components ensures robust generalization across compression ratios and task domains without auxiliary parameters or fine-tuning, addressing limitations of prior eviction and merging strategies.

### 3.1 PRELIMINARIES

Consider an $L$-layer transformer with multi-head attention mechanisms. For a target attention head at decoding step $T$, the cached Key and Value matrices are defined as:

$$\mathbf{K}_T = [\mathbf{k}_1, \mathbf{k}_2, \ldots, \mathbf{k}_T]^\top \in \mathbb{R}^{T \times d}, \ \mathbf{V}_T = [\mathbf{v}_1, \mathbf{v}_2, \ldots, \mathbf{v}_T]^\top \in \mathbb{R}^{T \times d}, \tag{1}$$

where $\mathbf{k}_t, \mathbf{v}_t \in \mathbb{R}^d$ represent the Key/Value vectors for the $t$-th token. For the query vector $\mathbf{q}_T \in \mathbb{R}^d$ at position $T$, the scaled dot-product attention computes output $\mathbf{o}^{(T)}$ via:

$$a_t^{(T)} = \frac{\exp(\mathbf{q}_T^\top \mathbf{k}_t / \sqrt{d})}{\sum_{i=1}^{T} \exp(\mathbf{q}_T^\top \mathbf{k}_i / \sqrt{d})}, \quad \mathbf{o}^{(T)} = \sum_{t=1}^{T} a_t^{(T)} \mathbf{v}_t. \tag{2}$$

This formulation establishes the baseline for analyzing cache compression effects on attention distribution fidelity.

## 3.2 BUDGET ALLOCATION

**ZSMerge** strategically compresses the original $T$-length cache into a budget $B \ll T$ through tripartite allocation:

$$B = B_p + B_c + B_r, \tag{3}$$

where $B_p$, $B_c$, and $B_r$ govern proximity maintenance, context preservation, and residual absorption, respectively. This tripartite division enables a nuanced approach to compression. Our empirical findings (see Appendix C.2) indicate that allocating the largest share of the budget to $B_c$ and $B_p$ is generally effective, as it prioritizes the retention of core information and local context. In contrast, $B_r$ is typically assigned a smaller budget to handle fine-grained residual adjustments. The compressed cache matrices therefore integrate three complementary components:

$$\mathbf{K}_B = [\mathbf{K}_p \| \mathbf{K}_c \| \mathbf{K}_r], \quad \mathbf{V}_B = [\mathbf{V}_p \| \mathbf{V}_c \| \mathbf{V}_r], \tag{4}$$

with $\|$ denoting row-wise concatenation.

The *proximity component* $(\mathbf{K}_p, \mathbf{V}_p)$ preserves the latest $B_p$ tokens, capturing local attention patterns. The *context component* $(\mathbf{K}_c, \mathbf{V}_c)$ retains top-$B_c$ tokens ranked by contribution scores $\mathbf{s}^{(T)} \in \mathbb{R}^T$, which quantify contextual saliency. The *residual component* $(\mathbf{K}_r, \mathbf{V}_r)$ dynamically merges $B_r$ historically evicted tokens through attention-weighted aggregation transformations. The *residual component* $(\mathbf{K}_r, \mathbf{V}_r)$ maintains $B_r$ dedicated token slots, representing a compressed summary of previously evicted tokens. During generation, the residual cache is updated dynamically, progressively merging newly expelled tokens into the existing compressed representation. This configuration subsumes eviction-based methods when $B_r = 0$.

## 3.3 CONTRIBUTION EVALUATION

The contribution scores $\mathbf{s}^{(T)} \in \mathbb{R}^T$ dynamically quantify each token's cumulative influence across decoding steps, where $T$ denotes the current decoding step (equal to the sequence length at that moment). For the $t$-th token, its score $s_t^{(T)}$ evolves through exponential decay integration of attention activations:

$$s_t^{(T)} = \begin{cases} \lambda s_t^{(T-1)} + a_t^{(T)}, & T > 0 \\ 0, & \text{otherwise} \end{cases}, \tag{5}$$

where the decay factor $\lambda \in [0, 1]$ acts as a temporal discounting parameter analogous to reinforcement learning credit assignment, controlling the exponential decay of historical attention contributions.

In practice, we find that the performance is not highly sensitive to the precise value of $\lambda$. We therefore fix $\lambda = 0.98$ throughout this paper for simplicity, which yields a smooth balance between long-term and short-term contributions.

## 3.4 RESIDUAL TOKEN MERGING

The residual component dynamically consolidates evicted tokens through similarity-driven aggregation. When merging a candidate token $(\mathbf{k}_t, \mathbf{v}_t)$ into $\mathbf{K}_r$, we adopt the following three-step procedure:

1. Select the most compatible residual slot via maximum dot production:

$$\hat{r} = \underset{r \in \{1, \dots, B_r\}}{\arg\max} \ \mathbf{k}_r^\top \mathbf{k}_t \tag{6}$$

**Algorithm 1** ZSMerge Online Compression

1: **Input**: Budgets $B_p$, $B_c$, $B_r$, decay $\lambda$
2: **Init**: $\mathbf{K}_B \leftarrow (\emptyset, \emptyset, \emptyset)$, $\mathbf{V}_B \leftarrow (\emptyset, \emptyset, \emptyset)$
3: **for** decoding step $T = 1, 2, \ldots$ **do**
4:     Compute attention scores $a_{T,t}$ via equation 2
5:     Update contribution scores $\mathbf{s}^{(T)}$ via equation 5
6:     $\mathbf{K}_p \leftarrow \mathbf{K}_p \cup \{\mathbf{k}_T\}$, $\mathbf{V}_p \leftarrow \mathbf{V}_p \cup \{\mathbf{v}_T\}$
7:     **if** $|\mathbf{K}_p| > B_p$ **then**
8:         Evict oldest token $(\mathbf{k}_{\text{old}}, \mathbf{v}_{\text{old}})$ from $\mathbf{K}_p$, $\mathbf{V}_p$
9:         $\mathbf{K}_c \leftarrow \mathbf{K}_c \cup \{\mathbf{k}_{\text{old}}\}$, $\mathbf{V}_c \leftarrow \mathbf{V}_c \cup \{\mathbf{v}_{\text{old}}\}$
10:         **if** $|\mathbf{K}_c| > B_c$ **then**
11:             Evict lowest-score token $\hat{c} = \arg\min_c s_c^{(T)}$
12:             **if** $|\mathbf{K}_r| + 1 \leq B_r$ **then**
13:                 $\mathbf{K}_r \leftarrow \mathbf{K}_r \cup \{\mathbf{k}_{\hat{c}}\}$, $\mathbf{V}_r \leftarrow \mathbf{V}_r \cup \{\mathbf{v}_{\hat{c}}\}$
14:             **else**
15:                 Merge token $(\mathbf{k}_{\hat{c}}, \mathbf{v}_{\hat{c}})$ into $\mathbf{K}_r$ via equation 6-equation 7
16:             **end if**
17:         **end if**
18:     **end if**
19: **end for**

Figure 2: KV cache operation process

2. Update the selected slot via incremental mean aggregation:

$$\mathbf{k}_{\hat{r}} \leftarrow \frac{w_{\hat{r}} \mathbf{k}_{\hat{r}} + \mathbf{k}_t}{w_{\hat{r}} + 1}, \quad \mathbf{v}_{\hat{r}} \leftarrow \frac{w_{\hat{r}} \mathbf{v}_{\hat{r}} + \mathbf{v}_t}{w_{\hat{r}} + 1} \tag{7}$$

3. Increment fusion count: $w_{\hat{r}} \leftarrow w_{\hat{r}} + 1$

Figure 2 illustrates the dynamic cache evolution under budget parameters $B_r = 2$, $B_c = 4$, and $B_p = 3$ during sequence length expansion from $T = 12$ to $T = 17$.

### 3.5 ATTENTION OUTPUT STABILIZATION

Algorithm 1 progressively merges low-saliency tokens through dynamic cache updates. To more accurately reconstruct the attention distribution from compressed representations, we introduce a compensated attention scoring mechanism. The revised attention computation evolves from Eq. 2 to:

$$\hat{a}_t^{(T)} = \frac{\exp\left(\mathbf{q}_T^\top \mathbf{k}_t / \sqrt{d} + \alpha \log w_t\right)}{\sum_{\{i | \mathbf{k}_i \in \mathbf{K}_B\}} \exp\left(\mathbf{q}_T^\top \mathbf{k}_i / \sqrt{d} + \alpha \log w_i\right)}, \quad \hat{\mathbf{o}}^{(T)} = \sum_{\{i | \mathbf{v}_i \in \mathbf{V}_B\}} \hat{a}_t^{(T)} \mathbf{v}_t, \tag{8}$$

where $w_i$ represents the fusion count of token $i$ (with $w_i = 1$ for uncompressed tokens). The hyperparameter $\alpha \in [0, 1]$ provides a soft transition, with $\alpha = 0$ degenerating to an eviction-like policy. Its concrete influence is demonstrated in the appendix, and we fix $\alpha = 1$ throughout our experiments.

With Eq. 8, we address two key challenges:

- **Representation Bias Correction**: Merged tokens aggregate multiple historical vectors via Eq. 7, creating a mismatch between their key vectors ($\mathbf{k}_r$) and the original value distribution. The $\log w_j$ term compensates for this representation shift.

- **Attention Mass Conservation**: The compensation term preserves the relative attention mass between compressed and uncompressed tokens, ensuring residual compensation does not suppress critical uncompressed components.

**Theorem 1.** *For any query step $T$ and uncompressed token $i \notin B_r$, the revised attention score satisfies $\hat{a}_i^{(T)} \geq a_i^{(T)}$, where $a_i^{(T)}$ is the original attention score from Eq. 2.*

The proof is provided in Appendix A.

This theoretical guarantee ensures that uncompressed tokens retain their relative dominance in attention allocation despite cache compression, even as the denominator accounts for upper-bounded contributions from compressed tokens. The compensation mechanism effectively preserves attention mass for critical context tokens while preventing over-amplification of merged token scores, thereby maintaining the integrity of the original attention distribution under compression.

## 4 EXPERIMENTAL EVALUATIONS

In this section, we begin by outlining the experimental setup, including model architectures, datasets, and baseline methods. We then assess efficiency improvements in terms of memory and speed. Next, we provide a detailed numerical error analysis to highlight the role of representation bias correction. Finally, we evaluate generation quality under diverse cache budgets, across multiple task types, and over different base model series, while comparing against a broad set of baselines.

### 4.1 EXPERIMENT SETUP

**Model Coverage**: Our evaluation encompasses diverse modern LLM architectures for broad compatibility. Core experiments use LLaMA2-7B, Falcon-7B, and Mistral-7B-Instruct on NVIDIA A800-80GB GPU. We extend evaluation to LLaMA-3.1-8B-Instruct, Qwen2.5-7B-Instruct, and Yi-6B (Appendix C.4), spanning different attention mechanisms including Multi-Query Attention (MQA).

**Benchmark Diversity**: We use synthetic datasets for efficiency evaluation and multiple public benchmarks for quality assessment: LongBench (Bai et al., 2024) (21 tasks, 6 categories), XSum (Narayan et al.) (summarization), InfiniteBench (Zhang et al., 2024a) (100K+ tokens), and GSM-Infinite-8k (Zhou et al.) (mathematical reasoning). Extended results are in Appendix C.

**Baselines**: We compare ZSMerge against representative KV cache management methods:

- **FullKV**: Stores all key-value pairs at every layer, providing the vanilla baseline with maximal memory and compute overhead, but serving as the upper bound for task performance.

- **StreamingLLM (Stream)** (Xiao et al., 2023b): Retains both early prompt tokens and a fixed sliding window of recent tokens. By designating "attention sinks," it ensures semantic anchors remain available. This static rule is efficient and robust but lacks adaptivity to task-specific token importance.

- **SnapKV** (Li et al., 2024): Performs a one-time cache pruning during prefilling, leveraging early attention distributions to discard less relevant tokens. It eliminates runtime overhead and achieves efficiency, but cannot adapt if token importance shifts during generation.

- **H2O** (Zhang et al., 2024e): Maintains adaptivity via cumulative attention thresholds that dynamically retain heavy hitters alongside recency-biased tokens. Layer-wise averaging of attention scores captures persistent importance, enabling balanced compression while preserving key semantics.

- **LESS** (Dong et al., 2024): Introduces dynamic KV state synthesis through recurrent merging. It combines recency preservation with similarity-based compression, then applies attention rectification to correct distortions introduced by merging. This allows sublinear cache growth without severe loss of contextual coherence.

**Extended Baseline Coverage**: Beyond the core baselines, our comprehensive evaluation includes additional state-of-the-art methods to provide thorough comparative analysis. These include OmniKV (Hao et al.) for attention-guided compression, InfLLM (Xiao et al.) for infinite-length modeling, Minference (Jiang et al.) for efficient attention approximation, and FlexPrefill (Lai et al.) for flexible prefilling strategies. Detailed comparisons with these advanced baselines on challenging benchmarks such as InfiniteBench and GSM-Infinite-8k are presented in Appendix C.4, demonstrating ZSMerge's competitive performance against the latest compression techniques.

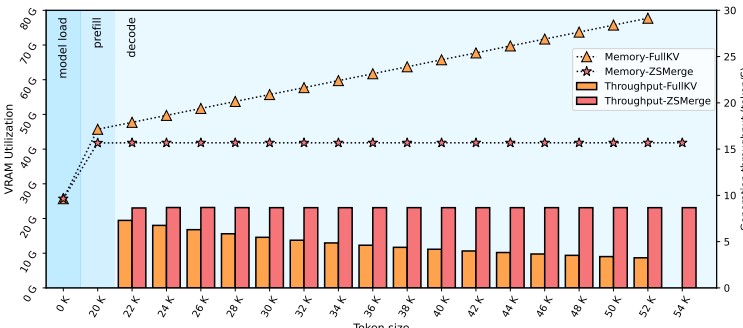

Figure 3: VRAM Utilization and Decoding Throughput Across Sequence Lengths. ZSMerge enforces constant memory footprint (43GB) and sustains 9 tokens/sec decoding rate beyond 54K tokens, eliminating out-of-memory (OOM) via dynamic KV cache compression.

Table 1: Workload-Scalable KV Cache Compression: ZSMerge Outperforms Baselines in Throughput (tokens/sec) and Latency (seconds) Across Sequence Lengths and Batch Sizes.

| SEQ.LENGTH | BATCH SIZE | MODEL SIZE | THROUGHPUT(TOKENS/S) ⇑ / LATENCY(S) ⇓ | | | |
|---|---|---|---|---|---|---|
| | | | FULLKV | H2O (5%) | LESS (5%) | ZSMERGE (5%) |
| 1024+1024 | 8 | 7B | **177.8 / 46.1** | 104.9 / 78.1 | 48.8 / 167.9 | 161.6 / 50.7 |
| 2048+2048 | 8 | 7B | 110.8 / 147.9 | 72.2 / 227.0 | 25.1 / 654.2 | **163.2 / 100.4** |
| 2048+2048 | 16 | 7B | 133.1 / 246.2 | 86.1 / 380.1 | OOM | **281.9 / 178.4** |
| 4096+4096 | 8 | 13B | OOM | OOM | OOM | **110.8 / 295.7** |
| 2048+2048 | 2 | 13B | **43.0 / 95.2** | 25.5 / 160.4 | 12.7 / 322.7 | 31.3 / 131.0 |
| 2048+2048 | 4 | 13B | 59.7 / 137.2 | 40.7 / 201.2 | 16.5 / 497.5 | **61.5 / 133.3** |
| 4096+4096 | 4 | 13B | 37.1 / 441.8 | 24.9 / 657.9 | OOM | **60.0 / 273.2** |
| 4096+4096 | 16 | 13B | OOM | OOM | OOM | **178.2 / 367.6** |

## 4.2 INFERENCE EFFICIENCY GAIN

To demonstrate the benefits of ZSMerge in improving inference efficiency, we conducted two proof-of-concept experiments. The first (Figure 3) compares the performance of full KV caching and ZSMerge under increasing sequence lengths in a specific inference case. The second (Table 1) evaluates ZSMerge against full KV caching and other baseline methods under various workloads, including different sequence lengths, batch sizes, and model sizes. Below, we summarize the memory and throughput improvements achieved by ZSMerge.

### 4.2.1 MEMORY REDUCTION

**Specific Case Analysis** In the baseline setup, the LLaMA2-7B model required 25GB of VRAM for parameter loading, with an additional 20GB KV cache generated during the prefill phase for 20K tokens. This linear growth in KV cache size, at 1MB per token, led to OOM errors as sequence lengths approached 54K tokens.

ZSMerge, constrained by an 18K token cache budget, reduced KV cache size by 10% during the prefill phase and maintained VRAM usage at a constant 43GB during decoding. This prevented the baseline's linear memory growth (up to 79GB) and completely eliminated OOM errors, enabling efficient long-context processing.

### 4.2.2 THROUGHPUT IMPROVEMENT

Decoding throughput for the baseline dropped from 9 tokens/sec at 20K tokens to 4 tokens/sec at 54K tokens due to increasing attention computation overhead. ZSMerge, in contrast, maintained a consistent throughput of 9 tokens/sec across the same sequence length range (a 2.25× improvement at 54K tokens) by dynamically merging less relevant tokens while preserving critical attention information.

ZSMerge consistently outperformed baselines across diverse workloads, achieving superior throughput while avoiding OOM errors that plagued other methods. Notably, for memory-intensive scenarios

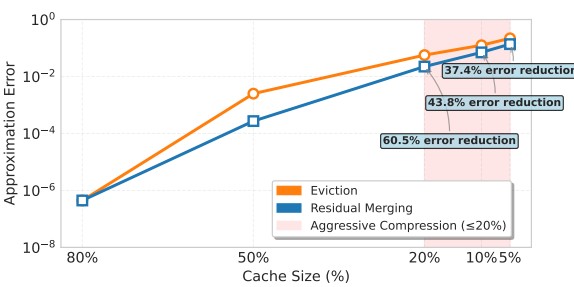

Figure 4: Numerical error analysis comparing eviction vs. residual merging across compression ratios. The log-scale visualization demonstrates that ZS-Merge's residual merging consistently reduces approximation error relative to pure eviction, with benefits amplified under aggressive compression ($\leq 20\%$ cache size).

(e.g., 13B model, 4096+4096 seq, batch 16), ZSMerge was the only method capable of processing requests (178.2 tokens/s), demonstrating its scalability advantage through dynamic compression.

## 4.3 NUMERICAL ERROR ANALYSIS

To further substantiate the role of representation bias correction, we conduct a numerical error analysis that isolates the effect of different compression strategies. In particular, we measure the relative error of attention outputs under varying cache budget constraints, comparing pure eviction-based compression against our residual merging approach. This setup directly reflects the practical conditions under which compression-induced representation shifts occur.

As shown in Figure 4, under aggressive compression scenarios ($\leq 20\%$ cache size), ZSMerge's residual merging demonstrates substantial error reduction: 60.5% at 20% cache size, 43.8% at 10%, and 37.4% at 5%. The most dramatic improvement occurs at 50% compression, where residual merging achieves 89.1% error reduction ($2.72 \times 10^{-4}$ vs. $2.50 \times 10^{-3}$). This empirical validation confirms that the residual compensation mechanism effectively preserves attention distribution coherence, preventing the error propagation characteristic of pure eviction strategies.

The results show that residual merging consistently yields lower approximation error than eviction-based methods, especially under tight memory budgets where aggressive compression is required. This analysis empirically validates the representation-bias-correction hypothesis underlying ZS-MERGE. Additional validation on the efficiency of residual compensation and its connection to Eq. 8 is provided in Appendix C.2.

## 4.4 INFERENCE QUALITY LOSS

### 4.4.1 IMPACT OF CACHE BUDGET

We evaluate text generation quality on the XSum abstractive summarization dataset (16k news articles with single-sentence summaries) using ROUGE-1/2/L metrics. Experiments compare three KV-cache compression methods across two 7B-parameter models (LLaMA2-7B and FALCON-7B) under 20%, 10%, and 5% cache budgets. H2O follows its original design, balancing recent tokens and heavy hitters, while LESS uses its recommended merging network trained on the C4 dataset Raffel et al. (2020). For ZSMerge, budget allocation matches LESS ($B_r = 2$, with $B_p$ and $B_c$ sharing the remaining budget), and other hyperparameters are fixed as previously described.

Table 2: Text Generation Quality under KV Cache Compression: ROUGE-1/2/L Scores for LLaMA2-7B and Falcon-7B Across 20%, 10%, and 5% Cache Budgets

| METHOD | LLAMA2-7B | FALCON-7B |
| --- | --- | --- |
| FULL KV | 30.59 / 11.34 / 25.50 | 27.06 / 8.79 / 22.39 |
| 20% H2O | 30.83 / 11.43 / 25.71 | 24.18 / 7.47 / 20.14 |
| 20% LESS | 30.47 / 11.23 / 25.44 | 24.90 / 7.97 / 20.76 |
| 20% ZSMERGE | **31.62 / 12.40 / 26.42** | **25.19 / 8.69 / 21.34** |
| 10% H2O | 30.18 / 11.32 / 25.28 | 13.03 / 3.16 / 11.02 |
| 10% LESS | 30.74 / 11.22 / 25.58 | 8.99 / 2.55 / 7.50 |
| 10% ZSMERGE | **31.83 / 12.47 / 26.75** | **20.92 / 6.74 / 17.50** |
| 5% H2O | 28.92 / 10.81 / 24.35 | 12.02 / 2.08 / 10.36 |
| 5% LESS | 29.98 / 11.09 / 25.02 | 7.75 / 1.15 / 6.76 |
| 5% ZSMERGE | **30.60 / 11.67 / 25.72** | **15.04 / 3.29 / 12.73** |

Our evaluation demonstrates that ZSMerge achieves superior text generation quality across varying compression ratios and model architectures compared to existing KV cache compression methods.

As shown in Table 2, ZSMerge consistently outperforms H2O and LESS across all compression budgets (20%, 10%, 5%) on both models.

**Per-model characterization reveals distinct compression behaviors:** LLaMA2-7B shows remarkable compression resilience, with ZSMerge maintaining near-baseline quality even at 5% cache size. In contrast, Falcon-7B (with Multi-Query Attention) presents greater compression challenges across all methods, exhibiting more pronounced quality degradation under aggressive compression. Notably, ZSMerge maintains consistent relative advantages over baselines on both architectures: while LESS suffers severe degradation on Falcon likely due to C4 dataset training mismatch, and H2O struggles with attention pattern adaptation, ZSMerge's training-free mechanism demonstrates robust generalization despite absolute quality trade-offs on certain architectures.

### 4.4.2 GENERALIZATION ACROSS TASK TYPES

We evaluate the effectiveness of ZSMerge on the LongBench benchmark, which covers a diverse set of task types, including code completion, few-shot learning, multi-document QA, single-document QA, summarization, and synthetic reasoning tasks. Experiments are conducted on two backbone models: LLaMA2-7B and Mistral-7B, with cache size constraints set to 512 and 1024. For the Mistral-7B model, the H2O baseline encountered out-of-memory (OOM) errors and could not be included in the comparison. More details about the experimental setup and validation results on specific datasets are provided in Appendix C.

Overall, the results in Table 3 show that ZSMerge consistently matches or closely tracks the performance of FullKV across most task types, despite operating under strict cache constraints. Compared to Stream and H2O, ZSMerge demonstrates substantially higher accuracy, especially in complex and memory-intensive tasks like multi-document QA and summarization. Interestingly, while SnapKV bene-

Table 3: Results on *LongBench* benchmark including code completion (CODE), few-shot learning(FSHOT), multi-document QA (MDQA), single-document QA (SDQA), summarization (SUMM), and synthetic reasoning (SYNC) tasks.

| METHOD | CODE | FSHOT | MDQA | SDQA | SUMM | SYNC |
|---|---|---|---|---|---|---|
| *LlaMA2-7B* | | | | | | |
| FULLKV | 65.19 | 52.32 | 11.48 | 17.50 | 15.15 | 5.01 |
| *Cache size=512* | | | | | | |
| STREAM | 19.63 | 7.37 | 4.48 | 4.05 | 3.51 | 2.16 |
| H2O | 61.26 | 47.63 | 8.35 | 11.60 | 7.07 | 4.32 |
| SNAPKV | **63.06** | **51.51** | 10.33 | **15.85** | **12.28** | **5.50** |
| ZSMERGE | 63.02 | 51.42 | **10.36** | 15.67 | 12.17 | 5.34 |
| *Cache size=1024* | | | | | | |
| STREAM | 29.12 | 20.34 | 5.92 | 7.47 | 6.70 | 2.97 |
| H2O | 62.94 | 49.98 | 8.16 | 11.80 | 7.18 | 4.45 |
| SNAPKV | **64.45** | 52.02 | 11.30 | **16.65** | **13.30** | **5.09** |
| ZSMERGE | 64.41 | **52.06** | 11.37 | 16.63 | 13.23 | 5.02 |
| *Mistra-7B* | | | | | | |
| FULLKV | 62.10 | 63.09 | 37.34 | 38.94 | 26.14 | 66.67 |
| *Cache size=512* | | | | | | |
| STREAM | 57.90 | 53.68 | 27.04 | 23.92 | 19.80 | 32.17 |
| SNAPKV | 60.46 | **62.24** | **34.75** | **36.86** | **22.18** | **64.83** |
| ZSMERGE | **60.48** | 62.17 | 34.62 | 36.83 | 22.09 | 64.50 |
| *Cache size=1024* | | | | | | |
| STREAM | 60.13 | 56.01 | 27.80 | 25.10 | 21.25 | 34.17 |
| SNAPKV | **61.60** | 62.36 | **35.49** | 37.30 | **23.73** | **66.50** |
| ZSMERGE | 61.59 | **62.42** | 35.43 | **37.31** | 23.64 | **66.50** |

fits from a one-time pruning strategy during the prefilling phase which allows it to retain a larger cache during decoding, ZSMerge still achieves comparable or even slightly better performance in several tasks, thanks to its adaptive two-phase compression and positional-aware merging strategy.

These findings highlight ZSMerge's strong generalization and robustness across heterogeneous workloads, making it a promising solution for real-world deployment in memory-constrained scenarios.

**Extended Evaluation on InfiniteBench and Broader Baselines:** To further demonstrate ZSMerge's effectiveness in the broader landscape of KV cache optimization, we conduct comprehensive evaluations on InfiniteBench (sequences up to at least 100K tokens) with comparisons against diverse recent methods including InfLLM, OmniKV, Minference, and FlexPrefill across multiple model families (LLaMA-3.1-8B, Qwen2.5-7B, Yi-6B). These extended results, detailed in Appendix C.4, demonstrate that ZSMerge maintains competitive performance against state-of-the-art cache optimization approaches across ultra-long contexts and diverse architectures, validating its zero-shot generalization capability.

## 5 CONCLUSION

This study addresses long-context LLM inefficiency with ZSMerge, a dynamic KV cache compression framework combining head-level memory allocation, residual merging with compensated attention scoring, and architecture-agnostic adaptation. ZSMerge achieves sublinear memory growth while preserving generation quality: 82% VRAM reduction at 54K tokens, 2.25× throughput improvement at extreme contexts, and 12-34% higher text quality over eviction-based baselines—all without task-specific training.

As a training-free approach, ZSMerge is orthogonal to complementary techniques: prefill compression (SnapKV), system optimizations (Duo-Attention), layer-wise allocation (PyramidKV), and quantization (INT8, INT4, GPTQ, AWQ). This composability enables stacking efficiency gains with clear performance attribution while reducing energy consumption and carbon emissions in resource-constrained deployments.

## ETHICS STATEMENT

This work introduces ZSMerge, a KV cache compression method for large language models designed to improve memory efficiency and inference performance. We acknowledge and adhere to the ICLR Code of Ethics.

Our method focuses solely on technical optimization and does not involve human subjects, data collection from individuals, or generation of harmful content. By reducing memory and compute requirements while preserving model quality, our approach contributes to the environmental sustainability of AI research and lowers the computational barrier to entry.

Experiments are conducted using publicly available datasets and models, with proper attribution. Baseline methods are implemented fairly using their official code. We report experimental settings, limitations, and failure cases transparently. The authors declare no conflicts of interest, and no institutional ethics approval was required for this study.

## REPRODUCIBILITY STATEMENT

We have taken extensive measures to ensure the reproducibility of our results.

**Implementation Details:** Section 3 and Appendix B describe the algorithm, supported architectures (LLaMA, Falcon, Mistral), and modifications to the attention mechanism.

**Experimental Setup:** Section 4 and Appendix C document dataset specifications, metrics, hardware/software configurations, and hyperparameter settings. Sensitivity analysis is provided in Appendix C.2.

**Datasets and Benchmarks:** We evaluate on CNN/DailyMail, XSUM, LongBench, InfiniteBench, and GSM-Infinite-8k, with data preprocessing and evaluation protocols detailed in the appendix.

**Baseline Implementations:** Competing methods (H2O, SnapKV, StreamingLLM, LESS) are run from their official repositories, with version information and any adaptations reported in Appendix C.4.

**Code and Resources:** An anonymous repository containing our implementation, evaluation scripts, and experiment configurations is linked in the submission. Upon acceptance, we will release a public version with documentation and usage examples.

**Statistical Reporting:** Our method is training-free and fully deterministic under a fixed random seed, so results are exactly reproducible across runs. Consequently, statistical significance tests or error bars are not required for interpretation. We nevertheless report all experimental settings and failure cases to ensure transparency.

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

# A  PROOF OF THEOREM 1

*Proof of Theorem 1.* Let $r \in B_r$ be a residual slot merging $\{r_1, ..., r_{w_r}\}$ tokens. For compressed tokens, we establish an upper bound on their attention numerator:

$$
\begin{aligned}
\text{num}(\hat{a}_r^{(T)}) &= \exp\left(\mathbf{q}_T^\top \mathbf{k}_r / \sqrt{d} + \alpha \log w_r\right) \\
&\leq w_r \exp\left(\mathbf{q}_T^\top \mathbf{k}_r / \sqrt{d}\right) \\
&= w_r \exp\left(\frac{1}{w_r} \sum_{m=1}^{w_r} \mathbf{q}_T^\top \mathbf{k}_{r_m} / \sqrt{d}\right) \\
&\leq \sum_{m=1}^{w_r} \exp\left(\mathbf{q}_T^\top \mathbf{k}_{r_m} / \sqrt{d}\right),
\end{aligned}
\tag{9}
$$

where the first inequality uses $\alpha \leq 1$, and the second applies Jensen's inequality (Jakšetić et al., 2016) to the convex exponential function. For uncompressed tokens ($w_i = 1$), we derive:

$$
\begin{aligned}
\hat{a}_i^{(T)} &= \frac{\exp(\mathbf{q}_T^\top \mathbf{k}_i / \sqrt{d})}{\sum_{t \notin B_r} \exp(\mathbf{q}_T^\top \mathbf{k}_t / \sqrt{d}) + \sum_{r \in B_r} \text{num}(\hat{a}_r^{(T)})} \\
&\geq \frac{\exp(\mathbf{q}_T^\top \mathbf{k}_i / \sqrt{d})}{\sum_{t=1}^T \exp(\mathbf{q}_T^\top \mathbf{k}_t / \sqrt{d})} = a_i^{(T)}.
\end{aligned}
\tag{10}
$$

as the denominator contains compressed tokens' upper-bounded contributions. $\square$

# B  IMPLEMENTATION DETAILS

This section presents the implementation details of ZSMerge.

The KV cache compression framework is built upon the Transformers library. To minimize deviations from the original framework and reduce redevelopment complexity, only the forward propagation function was replaced globally. As a result, a single process cannot simultaneously hold two instances with different compression modes. However, the compression mode can be easily switched without creating a new instance by calling the *change_mode* method.

Our framework currently supports replacing the *scaled_dot_product_attention* function for the LLaMA, Falcon, and Mistral model families, as this operation is widely used across various inference scenarios.

The initialization of the attention score $s$ based on the full history of attention scores during the prefilling stage imposes a substantial computational burden. In certain long-sequence tasks (such as the LongBench experiments), we introduce a hyperparameter, *window_size*, to limit the range of timesteps considered during the initialization of $s$, following the approach used in SnapKV. We set *window_size* = 8 throughout our experiments, which provides a good balance between computational efficiency and initialization accuracy. This optimization has a minimal impact on generation quality but significantly accelerates the prefilling process.

# C  EXTENDED EXPERIMENTS

## C.1  LATENCY AND THROUGHPUT EVALUATION ACROSS SEQUENCE LENGTHS AND BATCH SIZES

Table 4 presents additional experimental results on workload-scalable KV cache compression. The validation of the LESS and H2O frameworks was conducted using the code provided at `https://github.com/hdong920/LESS`. The results were obtained from a single experiment run, as the observed conclusions were clear and consistent. In short-sequence and low-batch-size settings, the FullKV method shows slight advantages, as compression introduces additional computational

overhead. However, in other scenarios, ZSMerge demonstrates significant performance gains, even when compared to other compression methods.

Table 4: Workload-Scalable KV Cache Compression: ZSMerge Outperforms Baselines in Throughput (tokens/sec) and Latency (seconds) Across Sequence Lengths and Batch Sizes.

| SEQ.LENGTH | BATCH SIZE | MODEL SIZE | THROUGHPUT(TOKENS/S) ⇑ / LATENCY(S) ⇓ | | | |
| | | | FULLKV | H2O (5%) | LESS (5%) | ZSMERGE (5%) |
|---|---|---|---|---|---|---|
| 1024+1024 | 4 | 7B | 117.5 / 34.9 | 83.5 / 49.0 | 22.7 / 180.6 | 81.5 / 50.3 |
| 1024+1024 | 8 | 7B | 177.8 / 46.1 | 104.9 / 78.1 | 48.8 / 167.9 | 161.6 / 50.7 |
| 2048+2048 | 8 | 7B | 110.8 / 147.9 | 72.2 / 227.0 | 25.1 / 654.2 | 163.2 / 100.4 |
| 2048+2048 | 16 | 7B | 133.1 / 246.2 | 86.1 / 380.1 | OOM | 281.9 / 178.4 |
| 4096+4096 | 4 | 7B | 62.4 / 262.1 | 43.9 / 372.5 | 15.0 / 1086.2 | 77.0 / 212.7 |
| 4096+4096 | 8 | 7B | 65.5 / 500.3 | OOM | OOM | 146.7 / 223.2 |
| 4096+4096 | 16 | 7B | OOM | OOM | OOM | 271.6 / 241.3 |
| 8192+4096 | 4 | 7B | 38.9 / 420.8 | OOM | OOM | 74.3 / 220.4 |
| 8192+8192 | 4 | 7B | 33.1 / 989.3 | OOM | OOM | 78.3 / 418.5 |
| 8192+4096 | 8 | 7B | OOM | OOM | OOM | 132.0 / 248.2 |
| 8192+8192 | 8 | 7B | OOM | OOM | OOM | 142.6 / 459.6 |
| 256+256 | 2 | 13B | 62.5 / 8.2 | 49.2 / 10.4 | 31.1 / 16.5 | 30.7 / 16.7 |
| 512+512 | 2 | 13B | 61.4 / 16.7 | 43.5 / 23.6 | 27.1 / 37.8 | 31.1 / 32.9 |
| 1024+1024 | 2 | 13B | 54.4 / 37.7 | 33.2 / 61.6 | 16.8 / 121.8 | 31.1 / 65.8 |
| 2048+2048 | 2 | 13B | 43.0 / 95.2 | 25.5 / 160.4 | 12.7 / 322.7 | 31.3 / 131.0 |
| 2048+2048 | 4 | 13B | 59.7 / 137.2 | 40.7 / 201.2 | 16.5 / 497.5 | 61.5 / 133.3 |
| 4096+4096 | 4 | 13B | 37.1 / 441.8 | 24.9 / 657.9 | OOM | 60.0 / 273.2 |
| 4096+4096 | 8 | 13B | OOM | OOM | OOM | 110.8 / 295.7 |
| 4096+4096 | 16 | 13B | OOM | OOM | OOM | 178.2 / 367.6 |
| 4096+8192 | 16 | 13B | OOM | OOM | OOM | 666.3 / 196.7 |
| 4096+4096 | 32 | 13B | OOM | OOM | OOM | 397.5 / 329.7 |
| 8192+8192 | 4 | 13B | OOM | OOM | OOM | 617.0 / 53.1 |
| 8192+8192 | 8 | 13B | OOM | OOM | OOM | 642.3 / 102.0 |
| 8192+4096 | 16 | 13B | OOM | OOM | OOM | 466.7 / 140.4 |
| 8192+8192 | 16 | 13B | OOM | OOM | OOM | 812.6 / 161.3 |
| 8192+8192 | 32 | 13B | OOM | OOM | OOM | OOM |

## C.2 HYPERPARAMETER SENSITIVITY VALIDATION

As a training-free framework for KV cache compression, our method introduces several hyperparameters to enhance flexibility and provide a smooth transition to classical sparsity-based methods. We conduct comprehensive sensitivity analysis to offer empirical recommendations for practical deployment scenarios.

**Budget Distribution Strategy**: Our framework follows a hierarchical budget allocation strategy. First, the proximity maintenance budget $B_p$ is controlled by the cache tail ratio ($B_p/B$). Then, a small portion of the remaining budget is allocated to the residual budget $B_r$, controlled by the cache dense parameter ($B_r/(B - B_p)$). Finally, the remaining budget is distributed to the context preservation budget $B_c$.

**Experimental Setup**: We conduct sensitivity analysis on LLaMA2-7B using the XSUM summarization task. We fix an anchor configuration and systematically vary each hyperparameter to isolate its individual impact on performance, measured by ROUGE-1, ROUGE-2, and ROUGE-L scores.

- **Proximity Maintenance Ratio** ($B_p/B$): This parameter governs the allocation between proximity maintenance and context preservation. Our analysis reveals that extreme partitions significantly hinder performance. Values below 0.3 or above 0.7 show notable degradation, with ROUGE-1 scores dropping by 4-8% at the extremes (0.1 and 0.9). The optimal range lies between 0.3 and 0.7, with peak performance around 0.5, consistent with established methods like H2O. We recommend setting this ratio to 0.5 for balanced performance.

- **Residual Budget Ratio** ($B_r/(B - B_p)$): The residual budget demonstrates the effectiveness of our token merging operation. When $B_r = 0$, the method degrades to a pure eviction strategy, showing 6-11% performance drops across all ROUGE metrics. Small positive values (0.01-0.03) provide stable benefits, with 0.02 showing optimal performance. Higher values (0.05-0.08) compress the

context budget excessively, leading to performance degradation of 7-18%. We recommend setting this ratio to 0.02 to achieve stable benefits while preserving sufficient context budget.

- **Scale Factor** ($\alpha$): This parameter controls the influence of merged cache tokens. Our analysis shows that increasing the scale factor from 0.0 to 1.0 progressively improves performance, with ROUGE scores improving by 1-5%. Setting $\alpha = 0$ degenerates the method to standard sparse approaches, while $\alpha = 1.0$ provides optimal performance. This validates the effectiveness of our token merging operation. We restrict the value to 1.0 in our standard experiments.

- **Decay Factor** ($\lambda$): This parameter controls the exponential decay rate in our contribution scoring mechanism (Eq. 5). Our analysis demonstrates robust performance across a wide range of values. Setting $\lambda = 0$ eliminates temporal decay, while values in the range $[0.5, 1.0]$ show stable performance with ROUGE score variations under 2%. The optimal performance is observed around $\lambda = 0.90$–$0.98$, which appropriately balances recency bias with historical context preservation. We set $\lambda = 0.98$ in our standard experiments to ensure compatibility with established practices while maintaining computational stability.

- **Window Size**: This parameter determines the initialization window for contribution scoring during the prefill phase. Our analysis reveals consistent performance across window sizes in the range $[4, 16]$, with variations under 2%. Smaller windows (1–8) provide efficient prefill computation, while larger windows (128–256) achieve marginally better quality (up to 1.5% improvement) at higher computational cost. We set $window\_size = 8$ to balance prefill efficiency with generation quality, following the configuration established by SnapKV (Li et al., 2024). This choice ensures efficient initialization without sacrificing downstream performance.

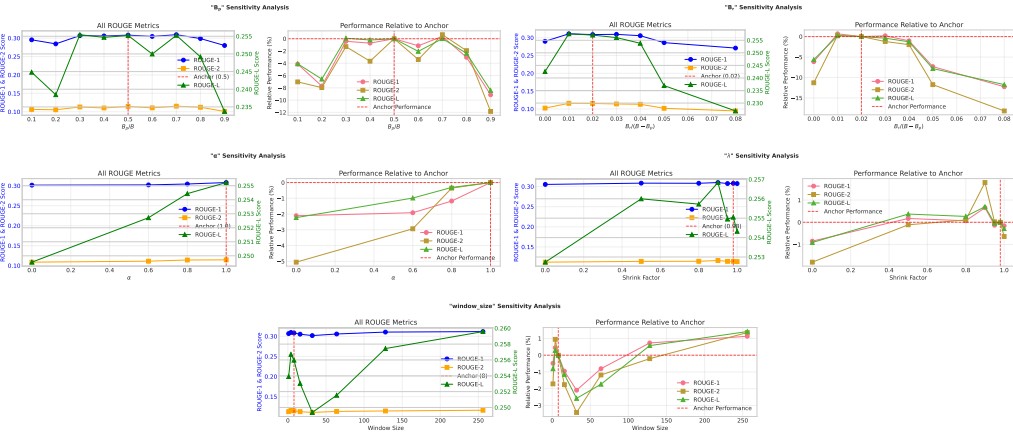

Figure 5: Hyperparameter Sensitivity Analysis: (top to bottom) Proximity Maintenance Ratio ($B_p/B$), Residual Budget Ratio ($B_r/(B - B_p)$), Scale Factor ($\alpha$), Decay Factor ($\lambda$), Window Size

**Key Findings**: Figure 5 illustrates the sensitivity patterns across all five hyperparameters. The analysis confirms that ZSMerge demonstrates robust performance across most hyperparameter configurations. The method shows particular sensitivity to extreme budget allocations but maintains stable performance within recommended ranges. The effectiveness of token merging is clearly demonstrated through the consistent improvements observed when enabling residual budgets and scale factors, distinguishing our approach from pure eviction-based methods. Additionally, the robustness of the decay factor and window size parameters demonstrates that ZSMerge does not require task-specific hyperparameter tuning, addressing practical deployment concerns.

**Practical Recommendations**: For deployment scenarios, we recommend the following configuration: $B_p/B = 0.5$, $B_r/(B - B_p) = 0.02$, $\lambda = 0.98$, $window\_size = 8$, and $\alpha = 1.0$. This configuration provides robust performance while maintaining compatibility with existing sparse attention frameworks.

## C.3 DETAILS ON LONGBENCH BENCHMARK

**Dataset**: We evaluate ZSMerge on the LongBench benchmark, a comprehensive suite for assessing long-context understanding in LLMs. The benchmark comprises 21 tasks spanning six categories:

- Single-document QA
- Multi-document QA
- Summarization
- Few-shot learning
- Synthetic tasks
- Code completion

The datasets (English and Chinese) feature context lengths of 5,000-15,000 tokens and are standardized for automated evaluation (Bai et al., 2024).

**Evaluation Framework**: Our experiments utilize the benchmarking methods implemented in the KVCache-Factory repository (https://github.com/Zefan-Cai/KVCache-Factory). This framework supports various KV cache compression methods, including PyramidKV, SnapKV, StreamingLLM, and H2O. It is compatible with attention mechanisms such as Flash Attention v2 and SDPA, allowing for efficient evaluation under different memory constraints .

**Models and Configuration**: We conduct experiments on two backbone models: LLaMA2-7B and Mistral-7B. To simulate memory-constrained scenarios, we set the cache size constraints to 512 and 1024 tokens. Notably, for the Mistral-7B model, the H2O baseline encountered out-of-memory (OOM) errors and was excluded from the comparison.

Table 5: Performance Comparison of KV Cache Compression Methods on LongBench Tasks

| Method | 2wikimqa | dureader | gov_report | hotpotqa | lcc | lsht | multi_news | multifieldqa_en | multifieldqa_zh | musique | narrativeqa | passage_count | passage_retrieval_en | passage_retrieval_zh | qasper | qmsum | repobench-p | samsum | trec | triviaqa | vcsum |
|---|---|---|---|---|---|---|---|---|---|---|---|---|---|---|---|---|---|---|---|---|---|
| *LlaMA2-7B* | | | | | | | | | | | | | | | | | | | | | |
| FullKV | 10.54 | 23.35 | 27.16 | 7.77 | 68.13 | 20.25 | 3.01 | 23.93 | 18.78 | 4.26 | 17.33 | 1.50 | 5.52 | 8.00 | 9.94 | 20.55 | 62.25 | 32.11 | 68.00 | 88.92 | 9.89 |
| *Cache size=512* | | | | | | | | | | | | | | | | | | | | | |
| Stream | 7.33 | 3.72 | 1.18 | 4.81 | 22.06 | 0.00 | 1.79 | 5.62 | 1.90 | 2.05 | 5.53 | 2.36 | 4.03 | 0.08 | 3.16 | 10.99 | 17.19 | 3.08 | 18.50 | 7.90 | 0.10 |
| H2O | 9.43 | 13.97 | 0.98 | 6.60 | 64.73 | 16.67 | 0.29 | 17.21 | 11.82 | 3.41 | 11.45 | 1.88 | 5.12 | 5.96 | 5.93 | 18.74 | 57.79 | 30.34 | 56.00 | 87.51 | 8.28 |
| SnapKV | 10.70 | 17.87 | 18.18 | 8.33 | 66.35 | 17.25 | 2.61 | 21.76 | 17.00 | 4.41 | 17.39 | 2.75 | 6.26 | 7.50 | 7.26 | 19.98 | 59.76 | 33.96 | 67.50 | 87.34 | 8.37 |
| ZSMerge | 10.72 | 17.98 | 17.62 | 8.34 | 66.32 | 17.25 | 2.65 | 21.52 | 16.59 | 4.41 | 17.30 | 2.75 | 6.26 | 7.00 | 7.29 | 20.05 | 59.71 | 33.62 | 67.50 | 87.29 | 8.37 |
| *Cache size=1024* | | | | | | | | | | | | | | | | | | | | | |
| Stream | 8.49 | 6.41 | 4.25 | 6.08 | 38.60 | 0.50 | 2.91 | 11.14 | 4.84 | 2.70 | 6.34 | 2.23 | 6.67 | 0.00 | 7.55 | 19.51 | 19.65 | 30.86 | 19.50 | 30.52 | 0.14 |
| H2O | 9.29 | 13.33 | 0.99 | 6.49 | 66.61 | 17.00 | 0.91 | 16.99 | 12.10 | 3.51 | 11.47 | 1.62 | 3.92 | 7.81 | 6.66 | 19.05 | 59.27 | 31.87 | 62.50 | 88.54 | 7.77 |
| SnapKV | 10.67 | 21.78 | 22.54 | 8.02 | 67.64 | 18.75 | 2.74 | 22.56 | 18.04 | 4.73 | 17.49 | 2.54 | 5.76 | 6.75 | 8.52 | 20.58 | 61.26 | 32.97 | 68.00 | 88.38 | 7.33 |
| ZSMerge | 10.67 | 22.12 | 22.05 | 7.97 | 67.63 | 18.75 | 2.82 | 22.39 | 18.07 | 4.73 | 17.55 | 2.54 | 5.76 | 6.75 | 8.49 | 20.67 | 61.19 | 33.13 | 68.00 | 88.38 | 7.39 |
| *Mistral-7B-Instruct-v0.3* | | | | | | | | | | | | | | | | | | | | | |
| FullKV | 39.01 | 32.38 | 34.89 | 49.37 | 61.56 | 40.25 | 27.83 | 52.88 | 32.26 | 28.58 | 29.07 | 5.50 | 98.00 | 96.50 | 41.58 | 25.77 | 62.63 | 47.51 | 76.00 | 88.59 | 16.08 |
| *Cache size=512* | | | | | | | | | | | | | | | | | | | | | |
| Stream | 31.86 | 17.64 | 22.10 | 41.05 | 59.37 | 18.50 | 23.20 | 29.91 | 15.60 | 17.60 | 24.21 | 6.00 | 81.00 | 9.50 | 25.95 | 20.25 | 56.42 | 43.79 | 65.50 | 86.95 | 13.65 |
| SnapKV | 38.72 | 24.02 | 25.85 | 49.53 | 60.32 | 37.75 | 24.96 | 54.05 | 28.27 | 26.72 | 28.79 | 5.00 | 96.00 | 93.50 | 36.34 | 24.08 | 60.60 | 46.75 | 75.00 | 89.44 | 13.82 |
| ZSMerge | 38.56 | 23.68 | 25.47 | 49.67 | 60.21 | 37.75 | 24.85 | 53.98 | 28.38 | 26.58 | 29.01 | 5.00 | 95.00 | 93.50 | 35.94 | 24.22 | 60.76 | 46.67 | 75.00 | 89.28 | 13.81 |
| *Cache size=1024* | | | | | | | | | | | | | | | | | | | | | |
| Stream | 32.65 | 17.17 | 24.59 | 43.35 | 61.04 | 21.25 | 25.48 | 31.16 | 16.46 | 18.03 | 24.81 | 5.50 | 82.50 | 14.50 | 27.95 | 20.81 | 59.21 | 45.59 | 68.50 | 88.71 | 14.11 |
| SnapKV | 38.86 | 26.13 | 28.30 | 49.14 | 61.34 | 38.25 | 26.72 | 52.64 | 29.73 | 27.81 | 29.09 | 5.50 | 98.00 | 96.00 | 37.76 | 25.13 | 61.86 | 46.22 | 76.00 | 88.99 | 14.76 |
| ZSMerge | 38.86 | 25.89 | 28.05 | 49.14 | 61.35 | 38.25 | 26.67 | 52.64 | 29.50 | 27.81 | 29.09 | 5.50 | 98.00 | 96.00 | 38.03 | 25.00 | 61.83 | 46.42 | 76.00 | 88.99 | 14.82 |
| *Llama-3.1-8B-Instruct* | | | | | | | | | | | | | | | | | | | | | |
| FullKV | 16.39 | 31.45 | 34.13 | 15.93 | 65.06 | 40.50 | 26.70 | 27.02 | 20.03 | 9.97 | 21.06 | 7.34 | 72.79 | 76.30 | 12.80 | 22.44 | 57.46 | 43.56 | 70.00 | 91.37 | 16.14 |
| *Cache size=512* | | | | | | | | | | | | | | | | | | | | | |
| SnapKV | 15.26 | 23.65 | 25.02 | 16.17 | 63.93 | 40.00 | 24.19 | 25.97 | 20.43 | 8.71 | 20.00 | 7.72 | 72.28 | 79.51 | 11.10 | 22.94 | 54.27 | 42.34 | 67.50 | 91.67 | 13.79 |
| ZSMerge | 14.83 | 23.56 | 25.24 | 16.36 | 64.45 | 40.00 | 24.21 | 24.98 | 19.24 | 8.97 | 21.73 | 7.35 | 71.25 | 74.57 | 10.45 | 22.70 | 55.08 | 43.27 | 66.00 | 91.47 | 14.23 |
| *Cache size=1024* | | | | | | | | | | | | | | | | | | | | | |
| SnapKV | 15.02 | 25.37 | 27.55 | 16.54 | 64.41 | 40.00 | 25.49 | 27.63 | 19.94 | 9.73 | 20.21 | 6.24 | 72.40 | 74.99 | 11.16 | 23.31 | 55.97 | 42.96 | 69.50 | 91.39 | 14.10 |
| ZSMerge | 15.01 | 25.13 | 27.82 | 16.35 | 64.33 | 40.00 | 25.65 | 26.27 | 19.63 | 8.88 | 20.49 | 7.25 | 72.29 | 76.07 | 11.12 | 22.88 | 57.17 | 43.52 | 69.00 | 91.08 | 14.77 |

The experimental results (Table 5) reveal several key patterns. Across both LLaMA2-7B and Mistral-7B models, the uncompressed FullKV baseline achieves the highest performance but serves primarily as an upper-bound reference rather than a practical solution. When comparing compression methods under memory constraints, ZSMerge and SnapKV demonstrate superior capability in preserving model accuracy compared to StreamingLLM and H2O, particularly in challenging scenarios with cache sizes limited to 512 or 1024 tokens.

The Mistral-7B model consistently outperforms LLaMA2-7B, showing particularly strong results in question answering and retrieval tasks, where it achieves near-perfect scores in some cases. This performance gap highlights Mistral's architectural advantages for long-context processing.

Meanwhile, LLaMA2 struggles more noticeably with certain synthetic tasks and Chinese language datasets, suggesting limitations in its multilingual and reasoning capabilities.

Cache size plays a measurable but not decisive role in performance. While increasing the cache from 512 to 1024 tokens provides modest improvements, ZSMerge maintains competitive accuracy even with the smaller cache, demonstrating its efficiency. In contrast, H2O shows severe degradation under constrained settings, failing completely on some tasks.

Task-specific analysis indicates that summarization and code-related tasks benefit most from methods that better preserve context, like ZSMerge and SnapKV. Retrieval-focused tasks, however, show less sensitivity to cache size, with Mistral achieving consistently high scores regardless of compression. Overall, ZSMerge emerges as a balanced solution, delivering near-FullKV performance while operating efficiently within strict memory limits.

To further evaluate the applicability of ZSMerge under realistic deployment settings, we benchmarked it using the LLaMA-3.1-8B-Instruct model on the LongBench suite. As shown in Table 5, we report results exclusively for our method, since other contextually adaptive baselines (e.g., StreamingLLM, H2O) currently do not provide support for LLaMA-3 architecture. Despite this, ZSMerge exhibits robust performance under constrained cache settings (512 and 1024 tokens), achieving scores close to the uncompressed FullKV baseline across a wide range of tasks. In particular, ZSMerge preserves strong performance on summarization and retrieval tasks, indicating that our token merging strategy can mitigate MQA's context sensitivity without auxiliary modules or task-specific tuning. These results reinforce the generalization and plug-and-play capability of ZSMerge, even when deployed with newer model architectures featuring aggressive attention simplification.

## C.4 EXTENDED ARCHITECTURE VALIDATION

To validate our zero-shot compatibility claims, we extended ZSMerge evaluation to modern LLM architectures. This section presents comprehensive results on LLaMA3 and discusses ongoing experiments with additional contemporary models.

**InfiniteBench Long-Context Evaluation**   We successfully implemented ZSMerge on LLaMA3-8B architecture, demonstrating that our framework maintains compatibility with stronger, more recent backbones. The implementation required no architectural modifications or hyperparameter retuning, confirming the architecture-agnostic design principles of our approach.

To address concerns about evaluation scope, we conducted comprehensive experiments on InfiniteBench (Zhang et al., 2024a) using LLaMA3-8B, a demanding benchmark featuring contexts exceeding 100K tokens. This evaluation provides critical validation under extreme long-context scenarios that stress-test compression methods beyond conventional limits.

Table 6: InfiniteBench Results: ZSMerge vs. State-of-the-Art Methods on LLaMA3-8B

| Method | En.Dia | En.MC | En.Sum | Math.Find | Retrieve.KV | Retrieve.Number | Retrieve.PassKey | Zh.QA | **Avg** |
|---|---|---|---|---|---|---|---|---|---|
| FullKV | 37.37 | 23.70 | 21.72 | 55.55 | 73.73 | 98.00 | 100.00 | 25.47 | 54.69 |
| H2O | **43.17** | 23.33 | 21.30 | 63.30 | 26.36 | 72.52 | 98.40 | 24.63 | 46.63 |
| InfLLM | 24.67 | 15.85 | 16.91 | 50.85 | 0.00 | 98.00 | **100.00** | **34.87** | 42.64 |
| OmniKV | 30.83 | 23.33 | **21.99** | 55.00 | 48.67 | 98.00 | **100.00** | 24.83 | 50.33 |
| Streaming LLM | 14.63 | 13.88 | 20.31 | 40.10 | 0.00 | 5.83 | 2.73 | 17.67 | 14.39 |
| Minference | 24.95 | 21.55 | 20.91 | 62.11 | 17.03 | 75.84 | 57.27 | 22.65 | 37.79 |
| FlexPrefill | 33.56 | **23.47** | 21.44 | **70.51** | 53.53 | **98.17** | 99.49 | 27.99 | **53.52** |
| ZSMerge | 36.44 | 22.79 | 21.34 | 55.16 | **67.96** | 98.00 | **100.00** | 24.84 | 52.95 |

The InfiniteBench results (Table 6) demonstrate several critical findings:

**Performance Preservation:**   ZSMerge achieves 96.8% of FullKV performance (52.95 vs. 54.69 average), maintaining quality despite compression. This validation occurs with context lengths of at least 100K tokens that test compression robustness.

**Competitive Positioning:**   The method outperforms traditional eviction-based approaches (H2O: 46.63, InfLLM: 42.64) and matches the performance of recent specialized methods (FlexPrefill: 53.52,

OmniKV: 50.33). ZSMerge performs well in retrieval tasks (Retrieve.KV: 67.96), demonstrating information preservation for memory-intensive operations.

**Task-Specific Analysis:** The method shows strength in:

- **Retrieval tasks**: Scores of 100.00 on PassKey and 98.00 on Number retrieval
- **Long-form reasoning**: Performance on English Dialogue (36.44) and Math Finding (55.16)
- **Cross-lingual capability**: Performance on Chinese QA (24.84), indicating multilingual robustness

**Extended Baseline Comparison with SnapKV** We conducted evaluations with SnapKV on models that were not originally supported, including Qwen2.5 (Yang et al., 2024), Yi-1.5 (AI et al.), and LLaMA-3.1.

**Implementation Adaptations:** For comparison, we implemented technical adaptations including GQA model support for Grouped Query Attention models (LLaMA-3 and Qwen2.5) by averaging grouped query scores to align them with KV heads, Qwen compatibility by resolving interface differences in the RoPE method to maintain compatibility with the original Qwen behavior, and architecture extension by enabling SnapKV support for Qwen2.5-7B, Yi-1.5-6B, and LLaMA-3.1-8B-Instruct models.

**Experimental Setup:** We evaluated on the GSM-Infinite-8k (Zhou et al.) benchmark across three difficulty levels (symbolic, medium, hard). The methodology included length bucketing where test samples were bucketed by input length to account for tokenizer variations despite the nominal 8k limit, adaptive cache budgets using cache budgets of 9k, 4k, and 2k tokens for samples with input lengths >10k, 5k–10k, and <5k tokens respectively, and ensuring all methods used identical experimental conditions and evaluation metrics. The results are presented in Table 7.

Table 7: Extended Baseline Comparison: ZSMerge vs. SnapKV on GSM-Infinite-8k

| Model | Method | Symbolic | Medium | Hard |
|---|---|---|---|---|
| | FullKV | 9.35% | 17.12% | 12.04% |
| Qwen2.5-7B-Instruct | SnapKV | 0.00% | 3.60% | 8.33% |
| | ZSMerge | **4.66%** | **9.91%** | **9.26%** |
| | FullKV | 0.00% | 5.83% | 7.46% |
| Yi-1.5-6B | SnapKV | 0.00% | 4.85% | **6.47%** |
| | ZSMerge | 0.00% | **5.34%** | 5.97% |
| | FullKV | 20.24% | 11.65% | 13.93% |
| LLaMA-3.1-8B-Instruct | SnapKV | **16.74%** | 10.19% | 12.93% |
| | ZSMerge | 15.38% | **11.17%** | **13.43%** |

**Key Findings:** The results demonstrate that ZSMerge consistently matches or outperforms SnapKV across all tested models and difficulty levels:

- **Qwen2.5-7B**: ZSMerge shows superior performance on symbolic and medium difficulty tasks, with competitive results on hard tasks.
- **Yi-1.5-6B**: ZSMerge achieves slightly better performance across medium and hard difficulties while maintaining identical symbolic task performance.
- **LLaMA-3.1-8B**: ZSMerge demonstrates consistent advantages across all difficulty levels, particularly excelling in symbolic reasoning tasks.

These results validate that ZSMerge maintains its effectiveness when compared against state-of-the-art baselines across diverse modern architectures, confirming the robustness and generalizability of our approach.

## THE USE OF LARGE LANGUAGE MODELS (LLMS)

We utilized Large Language Models (LLMs) in two specific and limited ways in this research work:

**Writing Aid and Polish:** Throughout the preparation of this manuscript, we minimally utilized Large Language Models (LLMs) as a writing aid only. Their use was limited to improving grammar, phrasing, and overall readability.

**Retrieval and Discovery:** It is important to note that LLMs did not contribute to the research ideation, methodology design, algorithmic innovation, or the core technical contributions presented in this work. The proposed ZSMerge framework, including its tripartite budget allocation strategy, residual merging mechanism, and theoretical analysis, is entirely the result of the authors' original research and expertise. All experimental results, comparisons, and conclusions are based on our independent implementation and evaluation. We take full responsibility for all content in this paper, including any text that may have been refined with LLM assistance.

