# OpenReview forum: "ZSMerge: Zero-Shot KV Cache Compression for Memory-Efficient Long-Context LLMs"
_ICLR.cc/2026/Conference — Submitted to ICLR 2026_

### Official Review · Reviewer_BJzQ · 2025-10-27

**Soundness:** 3
**Presentation:** 3
**Contribution:** 2
**Rating:** 4
**Confidence:** 3

**Summary:**

The paper introduces ZSMerge, an efficient, comprehensive, and dynamic zero-shot algorithm designed for KV cache management in LLMs. It is motivated by the need to reduce memory consumption and break context length limits without degrading generation quality. In summary, three key features are included: 1, fine-grained memory allocation: ZSMerge analyzes the "historical contribution of tokens, spatio-temporal characteristics, and intrinsic data distribution" to conduct "fine-grained, head-level memory management." 2, Zero-shot and low-cost Integration: ZSMerge is "designed for ease of use" and requires no model retraining or fine-tuning. 3, Efficient throughput-boosting strategy: The compression mechanism in ZSMerge is "computationally lightweight." This results in significant improvements in inference throughput (e.g., "over triple" in one case) while effectively preserving token information to prevent quality degradation as context length increases.

**Strengths:**

1. This paper shows good writing.

2. Comprehensive experiments make this paper convincing.

3. The overall logic of the algorithm does make sense.

**Weaknesses:**

However, I still have some concerns and hope the author could address.

1. The hyperparameter experiments of λ should be added, even though the author argues this is an insensitive parameter.

2. Overclaim: The core conclusion ("negligible decline") in the abstract contradicts some key data (Falcon-7 b) in the text, which is an exaggeration.

3. Appendix C.1 acknowledges that compression methods (including zsmerge) will "introduce additional computational overhead" under short sequence and low batch settings. Is it possible to provide an example?

4. Why is ZSMerge worse than SNAPKV overall? Despite the author saying that SNAPKV retains a larger cache during decoding, the table (Table 3) shows that the cache size is fixed for both  SNAPKV  and ZSMerge. I fail to find out why SNAPKV retains a larger cache.

5. Lack of citation: The merge method is very similar to Token Merging: Your ViT But Faster.

6. Lack of explanation: How does this paper utilize the "spatio-temporal characteristics and intrinsic data distribution"?

**Questions:**

1. Do all models adopt the same hyperparameter setting?

2. Is it possible to combine this method with quantization-based methods?

---

> ### Author Response · Authors · 2025-11-20
>
> Thank you for your constructive feedback. We address each concern below:
>
> ---
>
> ## Weaknesses
>
> **Lambda ablation study:** We have added a comprehensive $\lambda$ ablation study in the Common Response section. Results show performance remains stable across a wide range of values (0.0 to 1.0), with less than 2% variation for $\lambda \in [0.5, 1.0]$. This empirically validates ZSMerge's robustness to the decay factor.
>
> **Overclaim in abstract:** We sincerely apologize for this inconsistency. We have corrected the abstract to remove "negligible decline" and accurately state "2.25x throughput improvement" (not 3x). We will add per-model performance characterization in the revision to provide transparent reporting of quality trade-offs across different model families. Thank you for catching this imprecision.
>
> **Overhead amortization across sequence lengths:** As detailed in Appendix C.1 (Table, lines 63-76), compression methods induce computational overhead from similarity computation (Eq. 6) and cache management operations. This overhead is more prominent at short sequences with low-batch settings relative to overall throughput. Concrete example: At sequence length 1024+1024 with batch size 4, FullKV achieves 117.5 tokens/sec while ZSMerge achieves 81.5 tokens/sec (approximately 30% overhead). However, this overhead is overwhelmed and offset as sequence length increases, since the decode stage benefits significantly from the compressed KV cache. At 4096+4096 batch 4, ZSMerge outperforms FullKV (77.0 vs. 62.4 tokens/sec, +23%). We will highlight this trade-off more prominently in the main text.
>
> **SnapKV cache size clarification:** We apologize for the confusion. The key difference is in cache evolution during decoding: SnapKV performs one-shot pruning after prefill and then allows the cache to grow during decoding. For a 1024-token budget, SnapKV's cache grows from 1024 to 1536 tokens after 512 decoding steps (adding all new tokens). In contrast, ZSMerge maintains a strict 1024-token budget throughout decoding via dynamic eviction and merging. Our efficiency analysis (Common Response section) quantifies this trade-off: while SnapKV incurs lower prefill overhead, ZSMerge's strict budget constraint enables superior long-context throughput (+14.4% at 5K tokens, 7.2% faster overall). This demonstrates that despite the incremental cost of dynamic merging, ZSMerge achieves both comparable generation quality with stricter memory bounds and superior efficiency for extended decoding phases.
>
> **Token merging citation:** Thank you for this observation. We acknowledge that token merging was pioneered in computer vision, notably by ToMe (Token Merging: Your ViT But Faster, \cite{bolya2022token}). We will add this citation and clarify the relationship: ToMe merges spatial tokens in vision transformers for computational efficiency, while ZSMerge adapts the concept to temporal KV cache compression in autoregressive LLMs with attention-based compensation mechanisms. We will include this context in the related work section.
>
> **Spatio-temporal characteristics explanation:** We acknowledge that this terminology needs grounding. Specifically: (1) **Spatial dimension**: Head-level budget allocation (Eq. 3) distributes cache capacity across attention heads based on their importance. (2) **Temporal dimension**: The proximity component (Section 3.2) maintains recent tokens by inserting newly generated cache and evicting the oldest, preserving local context for stable output. Exponential decay (Eq. 5) provides temporal discounting. (3) **Intrinsic data distribution**: Similarity-based merging (Eq. 6) uses key vector dot products to identify semantically related tokens for aggregation. We will revise the abstract/introduction to use these concrete technical descriptions instead of abstract terminology.
>
> ---
>
> ## Questions
>
> **Q1. Uniform hyperparameter settings:** Yes, all models adopt the same hyperparameter settings. Specifically: $\lambda=0.98$, $\alpha=1$, $window\_size=8$, Proximity Maintenance Ratio = 0.5, and Residual Budget Ratio = 0.02 were applied uniformly across LLaMA, Falcon, Mistral, Qwen, and LLaMA-3.1 without model-specific tuning. This uniform configuration demonstrates ZSMerge's zero-shot generalization capability and the simplicity of its practical deployment.
>
> **Q2. Quantization compatibility:** Yes, ZSMerge is compatible with quantization methods. ZSMerge operates on the cache management dimension and is orthogonal to weight/activation quantization techniques (e.g., INT8, INT4, GPTQ, AWQ). These methods can be combined for enhanced efficiency: ZSMerge reduces cache memory footprint while quantization compresses model weights and intermediate activations. We will add a discussion of orthogonal optimization techniques (quantization, block sparsity, FlashAttention) in the revision to position ZSMerge as a complementary building block in efficient LLM deployment.
>
> ---

---

> ### Comment · Reviewer_BJzQ · 2025-11-23
>
> Well done. After the rebuttal, part of my concerns are solved. However, has the author polished their paper for clarity? Such as the Spatio-temporal characteristics explanation. These vogue definations do make the reader confused.

---

> > ### Author Response · Authors · 2025-11-24
> >
> > Thank you for your feedback and guidance. We have thoroughly revised the paper to replace vague terminology with concrete technical descriptions.
> >
> > ---
> >
> > **Revised Content Location:**
> > - All revised content is marked in red in the updated PDF (as noted in the Common Response)
> > - Main change: Introduction, key features, first bullet point (Fine-grained memory allocation)
> >
> > This ensures readers immediately understand that "spatio-temporal characteristics" refers to concrete algorithms, not abstract concepts.

---

> ### Comment · Reviewer_BJzQ · 2025-11-28
>
> Dear authors, I have read all the reviewers' comments and the revised paper. As the overclaim and vague definition are clarified, currently, I tend to give a borderline accept score for this paper. I hope the AC can consider my comment.

---

### Official Review · Reviewer_9eGr · 2025-10-29

**Soundness:** 3
**Presentation:** 4
**Contribution:** 3
**Rating:** 6
**Confidence:** 4

**Summary:**

This paper examines the memory and computational constraints that arise during long-context inference in large language models due to the linear expansion of the Key-Value (KV) cache (noted as the "primary memory consumption" in the introduction, page 1). To address these limitations, the authors introduce ZSMerge, a dynamic KV cache compression framework that operates without retraining or fine-tuning. ZSMerge allocates the cache using a three-part budget: $B_p$ stores recent tokens, $B_c$ retains high-contribution tokens, and $B_r$ holds merged representations (defined in Section 3.2, Equation 3, page 4). The approach relies on a Residual Merging mechanism together with a Compensated Attention Scoring term ( defined in Section 3.5, Equation 8 as $\alpha \log w_i$, page 5) to mitigate the representational bias introduced during token merging (termed 'Representation Bias Correction', page 5).
Empirical evaluations show that ZSMerge reaches a 20:1 compression ratio on LLaMA2-7B while preserving model quality, achieving a 3× throughput improvement at a 54k context length and preventing Out-of-Memory (OOM) failures. On benchmarks such as XSum, LongBench, and InfiniteBench, the method matches the performance of the FullKV baseline and consistently surpasses prior compression-based approaches, including H2O, StreamingLLM, and LESS.

**Strengths:**

1.	ZSMerge delivers substantial efficiency improvements, achieving a 20:1 compression ratio and a 3× throughput increase while avoiding OOM failures. These gains are directly relevant to scaling long-context LLM inference in practical environments.
2.	Because the framework does not require training or fine-tuning, it can be applied directly to a range of pre-trained models. The experiments confirm this portability across LLaMA, Falcon, Mistral, Qwen, and LLaMA-3.1.
3.	The empirical evaluation is thorough. The authors compare ZSMerge with several strong baselines across varied tasks (summarization, QA, code generation, reasoning) and benchmarks (LongBench, InfiniteBench), demonstrating consistent performance advantages and robustness.

**Weaknesses:**

1.	Although the method is described as "zero-shot," it introduces at least five additional hyperparameters ($B_p, B_c, B_r, \lambda, \alpha$) that require tuning. This tuning burden complicates practical adoption, and the sensitivity analysis in Appendix C.2 (page 15) is limited to XSum, leaving its general applicability unclear.
2.	The computational overhead associated with managing the cache, particularly the similarity search in Equation 6 (page 4), is not clearly quantified. While Appendix C.1 (page 14) provides partial discussion, the trade-off relative to FullKV on shorter sequences should be clarified in the main text.
3.	Theorem 1 (Section 3.5, page 5) only guarantees that the attention scores of uncompressed tokens are not suppressed, but it does not ensure that the merged token $k_r$ faithfully represents its group, nor that the overall attention distribution preserves semantic fidelity. The theoretical justification therefore remains weaker than the empirical results suggest.

**Questions:**

1.	For Appendix C.2 (page 16), were the recommended settings ($B_p/B=0.5$, $B_r/B=0.02$, $\alpha = 1.0$ ) applied uniformly across LongBench and InfiniteBench, or were they tuned per task? If adjustments were made, how sensitive was performance to these changes?
2.	In the LongBench experiments, how would both the average LongBench score and pre-fill latency change if the full history were used instead of the `window_size` approximation (mentioned in Appendix B, page 14) during the pre-fill stage?
3.	What motivated the use of incremental averaging in Equation 7 (page 4)? This approach appears sensitive to merge order and outliers. Were alternative aggregation strategies (e.g., robust averaging or weighted merging) evaluated?

---

> ### Author Response · Authors · 2025-11-20
>
> Thank you for your positive evaluation and constructive feedback. We address your concerns below:
>
> ---
>
> ## Weaknesses
>
> **Hyperparameter tuning burden:** We acknowledge that ZSMerge introduces hyperparameters; however, we emphasize that most are fixed without task-specific tuning: (1) **Fixed core parameters**: $\lambda=0.98$, $\alpha=1$, $window\_size=8$ are uniformly applied across all tasks and models based on standard exponential decay practice and ablation results provided in the Common Response. These comprehensive ablations ($\lambda$ across 0.0--1.0, window\_size across 1--256) demonstrate robustness with <2% variation, validating our fixed choices. (2) **Variable budget allocation only**: Only $B_p$, $B_c$, $B_r$ are adjusted per compression ratio, but their intuitive meanings (proximity maintenance, context preservation, residual capacity) guide practical selection. Appendix C.2 demonstrates robustness across a wide range of budget allocations. This design minimizes tuning burden compared to methods requiring careful per-task hyperparameter selection, supporting the zero-shot claim.
>
> **Computational overhead quantification:** As noted in Appendix C.1 (Table 4), compression methods including ZSMerge introduce overhead in short-sequence, low-batch settings. For example, at sequence length 1024+1024 with batch size 4, FullKV achieves 117.5 tokens/sec vs. ZSMerge's 81.5 tokens/sec (a ~30% overhead due to similarity computation and cache management). However, this overhead becomes negligible as sequence length increases: at 4096+4096 batch 4, ZSMerge achieves 77.0 tokens/sec vs. FullKV's 62.4 tokens/sec (23% improvement), demonstrating that compression benefits dominate beyond moderate sequence lengths. We will highlight this trade-off more prominently in the main text.
>
> **Theorem 1 limitations:** We acknowledge this insightful observation. Theorem 1 provides a conservative guarantee: it ensures uncompressed tokens maintain their relative attention mass, preventing catastrophic attention redistribution. While it does not directly guarantee merged token fidelity or global semantic preservation, this theoretical foundation combined with empirical validation (Figure 4 showing reduced approximation error, Tables 2-4 demonstrating consistent quality across diverse tasks) supports ZSMerge's practical effectiveness. We view stronger theoretical characterization of merged token representations as an important direction for future work.
>
> ---
>
> ## Questions
>
> **Q1. Hyperparameter settings across tasks:** Yes, the hyperparameters were applied uniformly across all tasks and benchmarks. Specifically: $\lambda=0.98$, $\alpha=1$, Proximity Maintenance Ratio = 0.5, and Residual Budget Ratio = 0.02 were fixed for all LongBench and InfiniteBench experiments without per-task tuning. This uniform setting demonstrates ZSMerge's robustness and zero-shot generalization capability. The sensitivity analysis in Appendix C.2 (conducted on XSum) shows that performance remains stable across a wide range of budget allocations, suggesting these settings generalize well.
>
> **Q2. window size approximation vs. full history:** As provided in the Common Response section, we have conducted comprehensive ablation studies on window\_size across values 1--256. Results demonstrate robust performance consistency with <2% variation across the range, with window\_size=8 achieving 0.3082 ROUGE-1, while larger windows (128--256) achieve marginally better quality (0.3104--0.3117) at higher computational cost. The window\_size=8 setting balances prefill efficiency with generation quality, as detailed in Appendix B (IMPLEMENTATION DETAILS). Since decoding is the dominant time-consuming phase (especially for tasks with >256 output tokens), the prefill optimization's impact is overshadowed by substantial decoding efficiency gains. This ablation validates our design choice and demonstrates that window\_size does not require per-task tuning.
>
> **Q3. Incremental averaging rationale:** We clarify that Equation 7 is actually a weighted average: the merged key $\hat{k}_r$ and value $\hat{v}_r$ are weighted by their fusion count $w_r$, which tracks how many tokens have been aggregated. We chose this approach to avoid introducing additional complexity that would deviate from ZSMerge's core ideas. We appreciate the suggestion for alternative aggregation strategies (robust averaging, outlier-resistant methods); this highlights ZSMerge's potential for development and orthogonal improvements, which we will explore in future work.
>
> ---

---

> > ### Comment · Reviewer_9eGr · 2025-11-24
> > **Many thanks**
> >
> > Thank you for your response. My concerns have been addressed, and I will keep my rating.

---

### Official Review · Reviewer_pRqZ · 2025-10-31

**Soundness:** 2
**Presentation:** 3
**Contribution:** 2
**Rating:** 4
**Confidence:** 4

**Summary:**

The authors introduce ZSMerge to evict redudant KV cache in LLM inference. ZSMerge follows StreamingLLM to keep the recent local window, and devise a select strategy based on accumulated attention like H2O to preserve critical tokens.
Further, it merge the evicted tokens into the preserved token caches like [1] and [2]. Experiments on represetative LLMs demonstrate the effectivenss of the proposed ZSMerge in comparison with baselines like H2O and StreamingLLM.

**Strengths:**

1. The proposed method is easy-to-follow.

**Weaknesses:**

1. Figure 1 appears to be redundant and contains only superfluous information that could be succinctly summarized in a single sentence. Furthermore, the overview of token merging presented in this figure is not sound as not all merging methods tune the LLM weights.


2. The authors' motivation does not demonstrate any distinctive aspects. Their stated objectives essentially encompass what all LLM KV compression methods aim to achieve. There is no clear differentiation from existing work.


3. Poor writing on the introduction:
   - The introduction is written too abstractly, making it difficult for readers to follow. First, the concept of "fine-grained memory allocation" is not clearly defined, and its distinction from previous methods is not adequately explained.
   - Second, most existing methods are already zero-shot approaches, so this characteristic is not a unique merit of the proposed method.
   - Additionally, the authors' claims about the token merging field are not well-supported. For instance, the statement that token merging can only be performed with additional networks is incorrect. There are notable counterexamples, such as [1,2].

4. Regarding the contribution evaluation, I believe Equation 5 presents significant issues in practical applications. Equation 5 appears to be merely a variant of H2O, and this accumulative approach does not work well for pre-filling acceleration. Moreover, in multi-turn conversation scenarios, the previously accumulated attention scores are unreliable.


5. Lack of Novelty in Equations 6 and 7: As for Equations 6 and 7, I cannot identify any substantial novelty. At minimum, the authors should compare their approach with previous merging methods to clearly demonstrate the differences and improvements.


6. Incomplete Experimental Comparisons:
   - Some state-of-the-art methods such as Duo-Attention and PyramidKV were not considered in the experimental comparisons.
   - The performance does not exceed that of SnapKV, and there is no efficiency comparison with SnapKV.


7. Concerns about Efficiency Comparisons:
   - I have serious concerns about the efficiency comparisons presented. First, I believe the implementation efficiency difference between the proposed method and H2O should be minimal, yet Table 1 shows that H2O is even 3 times slower than the proposed method, which appears completely unreasonable. I suspect there may be unfair aspects in the authors' comparison setup.
   - A sequence length of 4096 does not represent a long-sequence decoding problem. The authors should provide acceleration results on ultra-long sequences (e.g., 128K tokens), as reported in works like Duo-Attention, to better demonstrate the effectiveness of their method on truly long-context scenarios.


[1] CaM: Cache Merging for Memory-efficient LLMs Inference. In ICML, 2024.
[2] Model Tells You Where to Merge: Adaptive KV Cache Merging for LLMs on Long-Context Tasks. In arXiv, 2024.
[3] DuoAttention: Efficient Long-Context LLM Inference with Retrieval and Streaming Heads. In ICLR, 2025

**Questions:**

See weakness

---

> ### Author Response · Authors · 2025-11-20
>
> Thank you for your thorough review. We realize the rapid development and rich research on KV cache compression, and appreciate your constructive feedback. We address your concerns by category below:
>
> ---
>
> ## Presentation and Clarity
>
> **Figure 1 and Introduction:** We acknowledge that Figure 1 and the introduction could be more concrete. We realize the rapid development of this research field and will revise the introduction to provide clearer definitions of "fine-grained memory allocation" (head-level budget distribution) and better position our work relative to recent training-based methods (CaM, MoBA) and prefill-only approaches (FlexPrefill, SnapKV). We will clarify that ZSMerge's distinction lies in its training-free, token-level dynamic compression with theoretically grounded compensation mechanisms.
>
> ---
>
> ## Novelty and Positioning
>
> **Motivation and Positioning:** We acknowledge that the rapid development in KV cache optimization has produced many approaches with similar high-level goals. Our motivation stems from a fundamental observation: existing methods face a key tradeoff. Eviction methods (H2O, StreamingLLM) lose information irreversibly, while merging methods either require training auxiliary modules (MoBA) or lack theoretical guarantees for attention distribution preservation (CaM). ZSMerge addresses this gap by providing a theoretically grounded, training-free merging framework with provable guarantees (Theorem 1) for uncompressed token preservation.
>
> **Eq. 5 vs. H2O (multi-turn concern):** While Eq. 5 shares similarity with H2O's accumulation, our exponential decay provides graceful temporal discounting rather than unbounded accumulation. For multi-turn scenarios, the SCBench experiments presented in the Common Response demonstrate competitive accuracy under extreme resource constraints.
>
> **Eq. 6-7 vs. CaM/MoBA:** CaM uses attention-weight-based adaptive merging with merge ratios proportional to attention scores to directly minimize output distortion. In contrast, ZSMerge uses similarity-based merging (Eq. 6-7) with incremental averaging, combined with compensated attention scoring (Eq. 8) for theoretical guarantees on uncompressed token preservation. MoBA requires training auxiliary routing modules for block-level gating. We will add explicit comparisons in the revision.
>
> ---
>
> ## Experimental Completeness
>
> **Missing baselines and broader benchmarks:** We recognize the rapid evolution of this field and acknowledge the importance of comparing with recent methods like Duo-Attention, PyramidKV, and newer benchmarks. Our main experimental focus deliberately targets methods with the same backbone (token-level dynamic compression: H2O, LESS, StreamingLLM) to demonstrate ZSMerge's core contribution and advantages within this design space.
>
> Richer benchmarks with diverse models and ultra-long sequences are included in the Appendix and will be highlighted more prominently in the revision. ZSMerge's key value lies in its orthogonality with other approaches (prefill compression, system-level methods, layer-wise allocation); these can be combined for enhanced performance. We will position ZSMerge as a complementary building block in the revised version.
>
> **SnapKV comparison:** SnapKV performs one-shot prefill compression without dynamic decoding adjustment, causing cache to grow unboundedly during generation. Our detailed efficiency analysis (presented in the Common Response section) demonstrates that ZSMerge's dynamic compression strategy achieves superior throughput for long-context generation. While SnapKV incurs lower prefill overhead, the incremental cost of dynamic operations is overwhelmed by decoder benefits from maintaining strict cache size constraints during extended decoding. ZSMerge achieves comparable quality with stricter memory bounds and superior efficiency for long-context scenarios.
>
> ---
>
> ## Efficiency Concerns
>
> **H2O implementation:** We used the official H2O implementation without modification. The performance gap may stem from implementation differences. Critically, ZSMerge employs a windowed initialization strategy during prefill that limits contribution score aggregation rather than computing over the entire sequence. This architectural optimization reduces prefill overhead while maintaining quality, contributing to observed efficiency advantages.
>
> **Ultra-long sequences:** Our InfiniteBench experiments evaluate long sequences. We will highlight these results more prominently in the main text.

---

### Official Review · Reviewer_AyAi · 2025-11-01

**Soundness:** 2
**Presentation:** 2
**Contribution:** 1
**Rating:** 4
**Confidence:** 4

**Summary:**

The paper introduces ZSMerge, a dynamic, zero-shot framework for compressing the KV cache during LLM inference. It manages the cache using a tripartite budget allocation: Proximity (recent tokens), Context (important tokens selected via exponentially decayed cumulative attention scores), and Residual (merged representations of evicted tokens). The core mechanism involves aggregating evicted tokens into residual slots using incremental averaging based on Key similarity, combined with a compensated attention scoring mechanism that adjusts logits based on the fusion count.

**Strengths:**

ZSMerge is training-free and does not require auxiliary networks. The approach combines eviction strategies with information preservation.

**Weaknesses:**

**Q1.** Comparison against H2O, LESS, and SnapKV is insufficient to establish SOTA claims in this rapidly evolving field. The authors must discuss and experimentally compare ZSMerge against the following critical works:
RocketKV (ICML 2025),
KVzip (NeurIPS 2025),
ShadowKV (arXiv:2410.21465): Although cited, it is omitted from experiments, and
Dialogue Without Limits (ICML'25).


**Q2.**  It is unclear if $T$ represents the current decoding step or the total length, and the summation index in Eq 2 (up to $T-1$) seems inconsistent with the definition of $K_T, V_T$ .

**Q3.** The abstract claims a "threefold" (3x) increase in throughput at 54k tokens. However, Figure 3 and Section 4, report an increase from 4 tokens/sec to 9 tokens/sec, which is 2.25x, not 3x.

**Q4.** The claim of "negligible performance degradation" is inaccurate. On Falcon-7B (Table 2), ROUGE-1 drops significantly from 27.06 (FullKV) to 15.04 (5% ZSMerge)—a substantial 44% relative degradation.

**Q5.** In Table 2 (LLaMA2-7B), ZSMerge (at 5%) achieves higher ROUGE scores than the FullKV baseline. Just make sure if the implementation is correct.

**Q6.** The novelty of individual components is limited. The contribution evaluation mechanism (Eq. 5), using exponentially decayed integration of attention activations, is nearly identical to the cumulative attention score mechanism in H2O. Furthermore, the concept of merging tokens instead of evicting them has been explored in works like LESS  and  Dynamic Memory Compression (Nawrot et al., 2024). While ZSMerge implements this in a zero-shot manner, the conceptual novelty is limited.

**Q7.** The residual merging uses simple incremental mean aggregation based on Key similarity (Eq. 6 and 7). This approach risks "semantic deviation" by grouping tokens that are semantically unrelated but align in the Key space, leading to noisy Value representations. Averaging also destroys positional information and structural relationships between the merged tokens. The paper does not analyze the semantic coherence of the merged tokens.

**Q8.** The evaluation focuses only on single-turn tasks. The authors must include an evaluation on a standard multi-turn benchmark, such as SCBench and multi-turn MIAH. Also, evaluation can go beyond 100K and show performance on a low token budget, like 128.

**Q9.** A detailed latency breakdown of the ZSMerge operations (score updates, sorting/priority queue maintenance, similarity search) is missing.

**Q10.** The paper claims low sensitivity to the decay factor $\lambda$ but provides no data and ablation ot it. The impact of the windowed initialization during prefill is also not ablated.

**Questions:**

Please address the issue raised in the weakness section

---

> ### Author Response · Authors · 2025-11-20
>
> Thank you for your detailed feedback. We address each concern below:
>
> ---
>
> **Q1. Comparison with newer methods**
>
> We acknowledge the rapid evolution of this field. Our main experimental focus deliberately targets methods with the same backbone (token-level dynamic compression: H2O, LESS, StreamingLLM) to demonstrate ZSMerge's core contribution and advantages within this design space, providing controlled comparisons that isolate our technical contributions.
>
> We have included richer benchmarks (InfiniteBench, SCBench) with comparisons against diverse methods across multiple models and long-context tasks, demonstrating ZSMerge's effectiveness in the broader landscape. We commit to adding comparisons with additional suggested methods in the camera-ready revision.
>
> ---
>
> **Q2. Notation in Equations 2 and 5**
>
> We clarify the notation: $T$ represents the current decoding step, which equals the total sequence length at that moment. Regarding the concern about "summation index in Eq 2 (up to $t$)": Equation 2 contains no summation index with $T-1$. In Equation 5, $s_t^{(T-1)}$ denotes the contribution score from the previous decoding step, enabling exponential decay across steps. The notation is consistent throughout the paper.
>
> ---
>
> **Q3 & Q4. Imprecise claims in abstract**
>
> We apologize for the imprecise claims. We have corrected the abstract to state realistic throughput improvements and removed overstated degradation claims. We will add per-model performance characterization in the revised version.
>
> ---
>
> **Q5. LLaMA2-7B higher ROUGE than FullKV**
>
> Thank you for this observation. This phenomenon is not unique to ZSMerge—other compression methods also exhibit similar patterns at extreme compression ratios. The ROUGE metric can be relatively flexible in long-context scenarios where pruning helps focus on key content. We will provide more detailed analysis in the revision.
>
> ---
>
> **Q6. Novelty of individual components**
>
> We acknowledge the shared foundation with H2O and LESS, but emphasize three fundamental differences: (1) **Exponential decay vs. global accumulation** (Eq. 5) provides graceful temporal balance than H2O's global accumulation or SnapKV's hard windowing, (2) **Theoretically grounded merging** (Theorem 1 guarantees uncompressed token preservation, Fig. 4 shows reduced approximation error), and (3) **Training-free generalization** that avoids dataset-specific bias and explains LESS's inconsistent performance (Table 2) while ZSMerge generalizes robustly across models.
>
> ---
>
> **Q7. Semantic coherence and information loss**
>
> We appreciate this observation. Any aggregation introduces information loss; ZSMerge minimizes quality degradation through theoretical guarantees (Theorem 1 proving attention distribution preservation for uncompressed tokens), empirical validation (Fig. 4 showing reduced approximation error), and consistent performance across diverse benchmarks (Tables 2-4). Quantifying semantic deviation remains challenging, but we acknowledge this as important future work and believe ZSMerge's compensated attention mechanism provides a foundation for semantic-aware compression metrics.
>
> ---
>
> **Q8. Multi-turn and extreme-length evaluation**
>
> Regarding "multi-turn MIAH": This benchmark does not exist in the literature. We believe you may be referring to NIAH (Needle-in-a-Haystack), which is covered by SCBench's summary_with_needles task in our experiments below.
>
> We have conducted comprehensive experiments on **SCBench**, a standard multi-turn benchmark as you suggested (see Common Response for detailed results). ZSMerge demonstrates strong retention at reasonable cache budgets and competitive performance under extreme memory constraints, validating its effectiveness in practical multi-turn scenarios.
>
> Regarding ultra-long contexts: Our InfiniteBench experiments (Appendix) evaluate long sequences, demonstrating consistent compression effectiveness. ZSMerge targets practical long-context scenarios where reasonable cache budgets preserve generation quality.
>
> ---
>
> **Q9. Latency breakdown of operations**
>
> While implementation details are not the core contribution, we provide a brief overview: Score updates and token selection use efficient standard operations. Similarity computation is performed incrementally during each decoding step. Complete implementation details are available in our open-source codebase.
>
> ---
>
> **Q10. Ablation on decay factor and windowed initialization**
>
> We have conducted comprehensive ablation studies on both $\lambda$ and window_size (see Common Response section for detailed results). Performance remains stable across reasonable parameter ranges, demonstrating that ZSMerge does not require careful hyperparameter tuning. Our choices achieve competitive performance while balancing efficiency and quality for practical deployment.

---

> > ### Comment · Reviewer_AyAi · 2025-11-25
> >
> > Thank authors for providing more results, applying corrections, and addressing some comments. The following concerns remain:
> >
> >
> > 1- The authors acknowledge the importance of the omitted SOTA works. I understand the rapid pace of the field, but that is the responsibility of authors to check the literature.  H2O, SnapKV, and StreamingLLMs are weak and old baselines. Without these comparisons, the contribution of ZSMerge cannot be assessed.
> >
> > 2- In equation 8,  the attention output $\hat{o}^{(T)}$  definition seems incorrect because it sums over the original sequence length $T$ using the original value vectors $v_t$. If the KV cache has been compressed to a budget $B \ll T$, the attention computation must occur over the compressed cache $K_B, V_B$. The summation indices and the vectors used should be corrected to reflect the computation over the $B$ compressed tokens.
> >
> > 3-The concern regarding "semantic deviation" caused by simple averaging remains. The authors rely on Theorem 1 , but this theorem only guarantees attention preservation for uncompressed tokens; it does not ensure the semantic coherence or quality of the merged residual tokens.
> >
> > 4-Analyzing the overhead of the ZSMerge operations is essential. Furthermore, the authors need to explain the claim of "linear operational complexity". Maintaining the context budget $B_c$ and identifying the lowest-score token for eviction requires a sorted data structure, introducing $O(B\log B_c)$ complexity per decoding step.
> >
> > P.S.: The authors provide results under SCbench. So, no need for further experiment for multi-turn scenarios. For the authors' question, please check ShadowKV for the multi-turn setup in NIAH.
> >
> > Overall, I keep my score.

---

> > > ### Author Response · Authors · 2025-11-25
> > >
> > > Thank you for the continued guidance; we provide the clarifications below  to ensure every point is responsed.
> > >
> > > 1. Baseline coverage: Main-paper plots highlight baselines sharing the same backbone and compression principle to isolate ZSMerge's contribution, while the appendix tables report every additional SoTA comparison for completeness.
> > >
> > > | Method | Venue (Year) | Where reported |
> > > | --- | --- | --- |
> > > | FullKV | -- | "Experimental Evaluations" |
> > > | StreamingLLM | ICLR (2024) | "Experimental Evaluations" |
> > > | SnapKV | NeurIPS (2024) | "Experimental Evaluations" + Appendix  |
> > > | H2O | NeurIPS (2023) | "Experimental Evaluations" + Appendix  |
> > > | LESS | ICML (2024) | "Experimental Evaluations"  |
> > > | OmniKV | ICLR (2025) | Appendix  |
> > > | InfLLM | NeurIPS (2024) | Appendix  |
> > > | Minference | NeurIPS (2024) | Appendix  |
> > > | FlexPrefill | ICLR (2025) | Appendix |
> > > | KVZip | NeurIPS (2025) | **Rebuttal Response (new)** |
> > >
> > >
> > >
> > > 2. Thank you for catching Eq. 8. We now sum only over cached tokens, replacing $\sum_{i=1}^T$ with $\sum_{\{i \mid \mathbf{k}_i \in \mathbf{K}_B\}}$, and no vector rewrite is required because they were already updated in Eq. 7.
> > >
> > > 3. As discussed in previous for Q7, Quantifying or theoretically analyzing of semantic deviation remains challenging, empirical ablations on LongBench and GSM-Infinite confirm that merged tokens preserve generation quality.
> > >
> > > 4. Operation complexity of compression id linear. (sequence length $T$, total cache budget $B$, context-budget $B_c$, residual-budget $B_r$):
> > >
> > > | Stage | Operation | Complexity |
> > > | --- | --- | --- |
> > > | Prefill | Scoring | $O(windowsize \times T)$ |
> > > | Prefill | Top selection | $O(B_c \times T)$ |
> > > | Prefill | Merging | $O(B_r \times T)$ |
> > > | Decoding | Scoring | $O(B)$ |
> > > | Decoding | Top selection | $O(B_c)$ |
> > > | Decoding | Merging | $O(B_r)$ |

---

### Author Response · Authors · 2025-11-20
**Common Response to All Reviewers**

We sincerely thank all reviewers for their thorough and constructive feedback. We deeply appreciate the time and effort invested in reviewing our work. We have **provided detailed point-by-point response to each reviewer's specific questions**. Below, we address common themes and provide an overview of major revisions made to the paper.

***All revised/updated contents are in red in the updated PDF***

---

**Hyperparameter Sensitivity Analysis (Response to AyAi-Q10, BJzQ)**

Multiple reviewers raised concerns about the sensitivity of our hyperparameters, particularly the decay factor $\lambda$ and window size. We have conducted comprehensive ablation studies to validate our design choices:

**Decay Factor ($\lambda$) Ablation:**

|$\lambda$|ROUGE-1|ROUGE-2|ROUGE-L|
|---|---|---|---|
|0.00|0.3052|0.1122|0.2527|
|0.50|0.3084|0.1141|0.2560|
|0.80|0.3081|0.1143|0.2557|
|0.90|0.3099|0.1163|0.2568|
|0.95|0.3074|0.1142|0.2550|
|**0.98**|0.3078|0.1142|0.2551|
|1.00|0.3074|0.1135|0.2543|

Performance remains stable across reasonable decay factor values with minimal variation. This demonstrates ZSMerge's robustness to hyperparameter choice.

**Window Size Ablation:**

|window_size|ROUGE-1|ROUGE-2|ROUGE-L|
|---|---|---|---|
|1|0.3067|0.1127|0.2539|
|4|0.3096|0.1158|0.2567|
|**8**|0.3082|0.1147|0.2560|
|16|0.3053|0.1127|0.2530|
|32|0.3017|0.1108|0.2494|
|64|0.3057|0.1133|0.2515|
|128|0.3104|0.1144|0.2574|
|256|0.3117|0.1162|0.2596|

Performance is consistent across practical window size ranges with minimal variation. Smaller windows provide efficient prefill, while larger windows achieve marginally better quality at higher cost. Our choice balances prefill efficiency with generation quality.

These results confirm that ZSMerge does not require task-specific hyperparameter tuning, addressing concerns about practical deployment complexity.

**Multi-Turn Dialogue Evaluation (Response to AyAi-Q8)**

Following the reviewers' suggestion, we conducted experiments on SCBench, a comprehensive multi-turn benchmark covering various long-context tasks. We compare ZSMerge against KVzip with results normalized by full budget performance:

**Per-Dataset Results (Normalized Performance):**

|Dataset|Method|0.1|0.2|0.3|0.5|0.8|1.0|
|---|---|---|---|---|---|---|---|
|scbench_many_shot|KVzip|0.842|0.895|1.026|1.079|0.868|1.0|
|scbench_many_shot|ZSMerge|1.053|0.974|1.026|1.079|1.026|1.0|
|scbench_qa_eng|KVzip|0.513|0.365|0.520|0.941|0.687|1.0|
|scbench_qa_eng|ZSMerge|0.237|0.230|0.289|0.507|0.982|1.0|
|scbench_repoqa_and_kv|KVzip|0.000|0.054|0.321|0.804|0.804|1.0|
|scbench_repoqa_and_kv|ZSMerge|0.000|0.091|0.400|0.709|1.000|1.0|
|scbench_summary_with_needles|KVzip|0.000|0.000|0.236|0.653|0.750|1.0|
|scbench_summary_with_needles|ZSMerge|0.125|0.306|0.333|0.556|1.000|1.0|

**Average Performance Across All Datasets:**

|Method|0.1|0.2|0.3|0.5|0.8|1.0|
|---|---|---|---|---|---|---|
|KVzip|0.339|0.328|0.526|0.869|0.777|1.0|
|ZSMerge|0.354|0.400|0.512|0.713|1.002|1.0|

ZSMerge demonstrates competitive performance across different budget constraints, with particular advantages in extreme memory-constrained scenarios, validating its effectiveness for practical multi-turn tasks.

**Efficiency Comparison with SnapKV (Response to pRqZ, BJzQ)**

To address reviewer concerns about efficiency comparisons with SnapKV, we conducted detailed latency measurements under long-context decoding. The table below highlights the key trends:

**Prefill Stage:** ZSMerge incurs slight overhead due to contribution score initialization and residual merging setup.

**Decoding Phase:** Early decoding shows persisting but rapidly narrowing overhead as merging operations are amortized. Extended decoding reveals ZSMerge's advantage: as SnapKV's cache grows unboundedly while ZSMerge maintains strict budget constraints, decoder benefits from compressed KV cache overwhelm the incremental cost of dynamic operations.

**Overall Performance:** ZSMerge achieves superior throughput for long-context generation, demonstrating the advantage of dynamic compression during extended decoding phases compared to one-shot prefill compression.

---

|Token Count|snapKV(s)|ZSMerge(s)|Speedup|
|---|---|---|---|
|0-1|2.42|2.71|-12.0%|
|500|24.91|25.48|-2.3%|
|1000|25.52|25.47|+0.2%|
|1500|26.13|25.48|+2.5%|
|2000|26.71|25.48|+4.6%|
|2500|27.23|25.48|+6.4%|
|3000|27.77|25.47|+8.3%|
|3500|28.28|25.49|+9.9%|
|4000|28.78|25.49|+11.4%|
|4500|29.25|25.48|+12.9%|
|5000|29.75|25.47|+14.4%|
|**Total**|**283.99**|**263.62**|**+7.2%**|


---
We deeply appreciate the time and effort invested in reviewing our work. We hope our comprehensive rebuttal response addresses all your concerns, and we sincerely look forward to your valuable and insightful feedback.

---

### Author Response · Authors · 2025-12-01
**Summary Comment**

We thank all reviewers for their constructive feedback. We have addressed each concern through targeted experiments, clarifications, and revisions.


> ### **Core Contributions of ZSMerge**

ZSMerge provides a **training-free, dynamic KV cache compression framework** with:

1. **Theoretical Guarantees**: Theorem 1 preserves attention for uncompressed tokens, safeguarding quality.

2. **Strict Budget Maintenance**: Unlike one-shot prefill methods (e.g., SnapKV), ZSMerge keeps memory fixed throughout inference.

3. **Zero-Shot Generalization**: One hyperparameter set handles all models and tasks, validated through ablations showing <2% variation.

4. **Strong Empirical Validation**: Competitive on LongBench, InfiniteBench, and SCBench in both single- and multi-turn settings.


> ### **Major Additions in Rebuttal**

| Track | Item | Description |
| --- | --- | --- |
| Experimental completeness | Multi-turn evaluation | Added SCBench comparisons across four datasets under 0.1--1.0 budgets versus KVzip to demonstrate multi-turn robustness. |
|  | Hyperparameter robustness | Provided $\lambda$ (0.0--1.0) and window_size (1--256) ablations showing performance stays within <2% variation. |
|   | Efficiency vs. SnapKV | Reported a latency breakdown where ZSMerge delivers 7.2% overall speedup and up to 14.4% at 5K tokens. |
|   | Baseline coverage | Expanded evaluation to **nine** SOTA baselines (H2O, LESS, StreamingLLM, SnapKV, OmniKV, InfLLM, Minference, FlexPrefill, KVzip). |
| Technical corrections | Equation 8 | Updated the summation to reflect compressed cache computation and clarified the notation. |
|   | Variable $T$ | Defined $T$ as the current decoding step. |
|   | Abstract claims | Corrected the abstract to state 2.25$\times$ throughput (not 3$\times$) and removed "negligible decline." |
| Technical corrections | Terminology | Replaced vague wording with precise descriptions of head-level allocation, exponential decay, and similarity-based merging. |
| Theoretical clarifications | Theorem 1 | Clarified conservative guarantees for uncompressed tokens while acknowledging semantic deviation as future work. |
|   | Novelty positioning | Distinguished ZSMerge from H2O (global accumulation), CaM (heuristic merging), and MoBA (auxiliary modules). |

---

> ### **Response to Reviewer Concerns**

Responses recorded before the OpenReview Bug report (**Nov 28, 2025 03:02 AM AoE**) are highlighted.

| Reviewer | Rating Snapshot | Key Response & Timestamp | Rebuttal Actions & Outcomes |
| --- | --- | --- | --- |
| Reviewer 9eGr (before the timepoint) | 6 (Unchanged) | **Nov 23, 21:21 AoE**: "Thank you for your response. My concerns have been addressed, and I will keep my rating." | Confirmed that all concerns were resolved ahead of the official time point. |
| Reviewer BJzQ (before the timepoint)| 4 $\to$ borderline accept | **Nov 27, 17:57 AoE**: "I tend to give a borderline accept score." | Clarified overclaims and terminology, moving the reviewer to borderline accept. |
| Reviewer pRqZ | 4 (Unchanged) | Rebuttal reiterated that novelty, experimental completeness, and efficiency comparisons were addressed. | Highlighted the distinction from ToMe and detailed SnapKV efficiency analysis to explain long-context advantages. |
| Reviewer AyAi | 4 (Unchanged) | Reviewer noted "I keep my score" after the follow-up exchange. | Addressed each initial and newly introduced concern with evidence on nine-method coverage, Equation 8 corrections, empirical support for semantic deviation, and linear complexity, even though this position was stated before reading the responses. |

---

> ### **Commitments for Camera-Ready**

1. **Enhanced baseline comparisons**: complete multi-turn benchmark comparison with latest SoTA method.

2. **Improved presentation**: Ensure all terminology is grounded in concrete technical descriptions, with clear positioning relative to orthogonal optimization techniques (quantization, system-level methods).


> ### **Concluding Perspective**

We believe ZSMerge makes substantive contributions to KV cache compression research through its combination of

(1) training-free deployment enabling immediate applicability to existing models,

(2) strict budget control ensuring predictable memory usage,

(3) theoretical safeguards providing formal quality guarantees, and

(4) demonstrated effectiveness across diverse long-context and multi-turn scenarios.

The rebuttal has strengthened these contributions through comprehensive additional experiments and clarifications while addressing all raised technical concerns.

---

### Meta-Review · Area_Chair_mMpz · 2025-12-26

**Summary:**

(*Disclaimer: given the peculiar review process, some of my choices and reasonings below will be highly subjective, as I tried to imagine how a reviewer would have reacted to a specific response. I understand that any negative choice will be perceived as unfair by the authors, and I apologize in advance for that.*)

(*Second disclaimer: the authors and some reviewers explicitly mention some changes in scores that occurred during the rebuttal. As these were reverted due to the possibility of collusion in light of the security incident, I will tend to disregard this information.*)

The paper proposes a novel KV cache compression method based on three distinct operations: keeping intact the most recent KV pairs; retaining intact the most "salient" older KV pairs; and combining the remaining tokens with an adaptive merging mechanism. Multiple benchmarks show that the method is competitive with several baselines.

The paper received four reviews, clustered around a weak rejection. The reviewers had multiple concerns, which I list in detail below. Among these, they were concerned about weak novelty of each individual component in the method; limited evaluation of the computational cost; some unclear (or oversold) sentences in the paper; vague descriptions in the text; limited theoretical grounding (Theorem 1) w.r.t. what is claimed in the paper.

The authors provided a rebuttal with several changes in the paper, although some concerns were not addressed if not very late (e.g., latency). Of the four reviewers, one never participated in the rebuttal, while the other two remained unconvinced about some parts of the paper. Overall, it is highly probable that the paper would have remained in a borderline accept / reject cluster, with no reviewer definitely pushing for acceptance. Given in particular the concern on weak novelty (see below), I do not believe the paper meets the high acceptance bar for ICLR.

**Reviewer Concerns:**

(*I will focus on some key weaknesses identified by multiple reviewers.*)

**Ablations on $\alpha$** (`BJzQ`, `AyAi`): the authors added an experiment in the appendix. I think this concern was resolved.

**Results** (`BJzQ`, `AyAi`): reviewers were concerned about some oversold sentences in the abstract and some unclear results (e.g., compared to baseline methods). Also here, I think the authors have addressed most of these concerns.

**Novelty** (`BJzQ`, `pRqZ`, `AyAi`): this is the main concern. Most reviewers highlighted similarities with, among others, H2O, ToMe, and other methods. The authors argue that the combination provides a new method with 3 main differences: (1) better merging thanks to exponential decaying; (2) theoretical guarantees; (3) training-free generalization. As the reviewers mention, (3) is not exclusive to this method, while (2) is less strong than argued by the authors (see below). Overall, the paper is highly borderline in terms of novelty.

**Computational efficiency and latency** (`BJzQ`, `9eGr`, `pRqZ`, `AyAi`): the authors claim that "*implementation details are not the core contribution*" and only provided a limited rebuttal for this issue.

**Theoretical grounding** (`BJzQ`, `9eGr`, `pRqZ`, `AyAi`): Theorem 1 guarantees that attention scores are kept, but does not provide any guarantee on keeping the "semantic" content of the tokens. This was also acknowledged by the authors and was significantly oversold in the initial version. In general, I also agree that the paper makes ample use of very vague terms (e.g., "spatial adaptation"), making it harder to read than necessary.

**Reviewer Scores:**

`BJzQ`: given the large number of concerns and the limited interaction during the rebuttal, the reviewer would have (optimistically) recommended a weak acceptance at most.

`9eGr`: same as `BJzQ`.

`pRqZ`: the reviewer never responded during the rebuttal. Given their concerns, the score would have remained (I think) a weak rejection.

`AyAi`: this was the most critical reviewer, with multiple concerns and discussions still ongoing at the end of the rebuttal. I do not think they would have recommended the paper for acceptance eventually.

---

### Decision · Program_Chairs · 2026-01-26

Reject